# CoBel-World: Harnessing LLM Reasoning to Build a Collaborative Belief World for Optimizing Embodied Multi-Agent Collaboration

## Abstract

Effective real-world multi-agent collaboration requires not only accurate planning but also the ability to reason about collaborators' intents—a crucial capability for avoiding miscoordination and redundant communication under partial observable environments. Due to their strong planning and reasoning capabilities, large language models (LLMs) have emerged as promising autonomous agents for collaborative task solving. However, existing collaboration frameworks for LLMs overlook their reasoning potential for *dynamic intent inference*, and thus produce inconsistent plans and redundant communication, reducing collaboration efficiency. To bridge this gap, we propose **CoBel-World**, a novel framework that equips LLM agents with a **collaborative belief world**—an internal representation jointly modeling the physical environment and collaborators' mental states. CoBel-World enables agents to parse open-world task knowledge into structured beliefs via a symbolic belief language, and perform zero-shot Bayesian-style belief updates through LLM reasoning. This allows agents to proactively detect potential miscoordination (e.g., conflicting plans) and communicate adaptively. Evaluated on challenging embodied benchmarks (i.e., TDW-MAT and C-WAH), CoBel-World significantly reduces communication costs by **22-60%** and improves task completion efficiency by **4-28%** compared to the strongest baseline. Our results show that explicit, intent-aware belief modeling is essential for efficient and human-like collaboration in LLM-based multi-agent systems.

## 1 Introduction

Collaboration is a fundamental social mechanism through which humans solve complex tasks and reshape their environments. In recent years, large language models (LLMs) have demonstrated remarkable capabilities in reasoning, planning, and decision-making (Liu et al., 2024a; OpenAI, 2023; Comanici et al., 2025; Wu et al., 2025), suggesting growing potential for LLMs to act as autonomous agents capable of participating in collaborative problem-solving. While these advances are promising, the effectiveness of existing LLM-based collaboration frameworks has largely been confined to simple, text-based domains with high environmental certainty (Hong et al., 2023). In contrast, real-world collaboration requires agents to coordinate actions under uncertainty and adapt to dynamic, partially observable environments. This raises a key question: Can LLMs, when grounded in the physical world, autonomously coordinate with other agents for effective and efficient collaboration?

We investigate this question in the context of decentralized embodied multi-agent tasks (Zhang et al., 2023; Nayak et al., 2024; Kannan et al., 2023), where agents must perceive, plan, and act under partial observation (Spaan et al., 2006b; Bernstein et al., 2002), long-horizon dependencies, and environmental dynamics. In such settings, the primary challenge stems from incomplete and misaligned information across agents (Bernstein et al., 2002; Foerster et al., 2019). Communication thus becomes essential for synchronizing internal states, sharing observations, and aligning intents.

As shown in Figure 1, recent approaches have explored various communication protocols to enable information sharing and consensus in multi-agent systems. However, these methods typically rely on predefined collaboration or schemes and fixed communication protocols-such as step-by-step message generation (Zhang et al., 2023), dense discussion (Mandi et al., 2024), or event-triggered

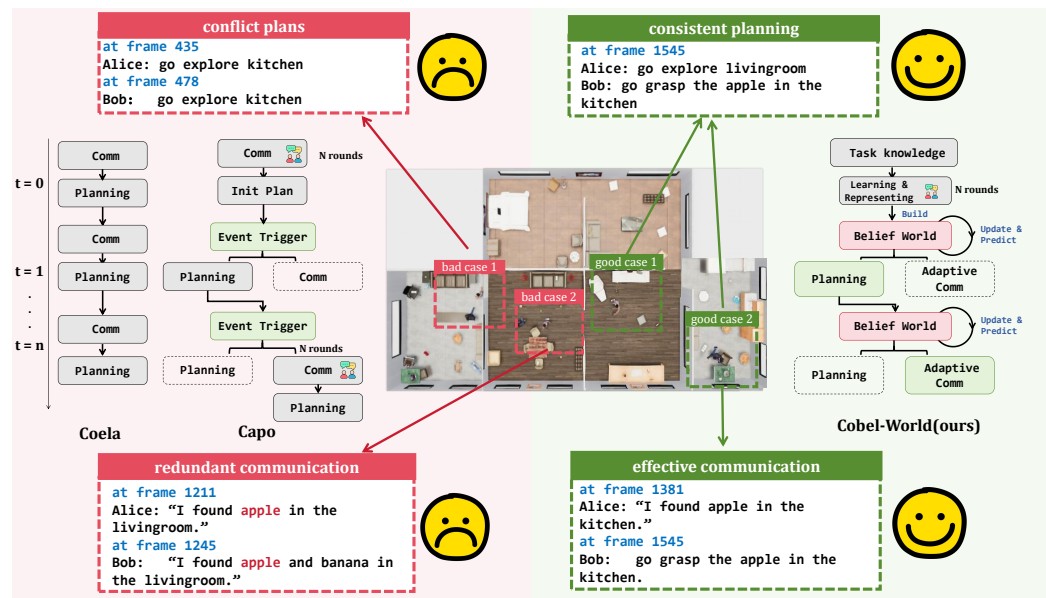

Figure 1: **Comparison of existing works with our work.** From left to right: (a) CoELA (Zhang et al., 2023): A collaboration framework based on step-by-step templated message generation and planning. (b) Capo (Liu et al., 2024b): A collaboration framework based on event-driven multi-round discussion. (c) Our CoBel-World framework, featuring belief modeling and adaptive collaboration. Our method enables consistent planning and effective communication.

multi-round discussion (Liu et al., 2024b). Crucially, they lack the ability to dynamically identify potential miscoordination and communicate adpatively. As a result, redundant communication and inconsistent planning are common, leading to heavy communication costs and redundant physical actions. These limitations hinder scalability in large-scale, communication-constrained, or human-AI collaborative environments.

We argue that this shortcoming arises from the absence of belief modeling. In multi-agent systems, beliefs refer to an agent's internal representation of possible states—including the external environment and the mental states (e.g., intents, knowledge) of collaborators (Kominis & Geffner, 2015; Geffner & Bonet, 2013). In decentralized multi-agent reinforcement learning (DEC-MARL), belief modeling has proven critical for collaboration under partial observation, enabling agents to infer and align with others' internal states or policies (Pritz & Leung, 2025; Wen et al., 2019; Zhai et al., 2023). With accurate belief estimation, agents can selectively communicate only the valuable information to achieve efficient communication and reach consensus, thus promoting consistent collaboration.

Despite its advantages, modeling belief for LLM-driven agents presents two primary challenges:

1. **Challenge 1: Formulating beliefs in open-ended environments.** Traditional MARL agents operate in low-dimensional, structured environments (e.g., grid worlds) with discrete action space, enabling straightforward belief representation. In contrast, LLM-based embodied agents interact with open-ended physical environments characterized by high-dimensional, compositional actions, and free-form communication. These features complicate the grounding of linguistic instructions into structured, explicit belief representation.

2. **Challenge 2: Zero-shot construction of belief models.** In abstract domains like grid-world games (Moreno et al., 2021), agents are trained on large-scale interaction datasets to infer others' intents. However, collecting real-world interaction trajectories for fine-tuning LLM agents is prohibitively expensive and often impractical. Moreover, data-driven models may struggle to generalize across diverse, unseen scenarios. This necessitates a zero-shot approach: LLM agents must construct and update beliefs without access to annotated interaction data during pretraining or downstream adaptation.

To address these challenges, we propose ***CoBel-World***, a novel framework equips LLM agents with a ***collaborative bel***ief world—an internal representation of the external world and mental states of collaborators. We leverage the advanced reasoning capabilities of LLMs to predict possible beliefs based on observed information, thereby bridging the gap caused by the lack of collaborative data during pretraining. This model enables agents to reason about the internal states of collaborators and predict the future states of the external world, facilitating more efficient and human-like collaboration. Specifically, CoBel-World incorporates two core components. First, inspired by symbolic planning languages such as PDDL (Fox & Long, 2003; Fabiano et al., 2021), we introduce a symbolic belief language to formalize the multi-agent task settings. Then, the agents will learn knowledge about the external world and represent it as belief rules to guide subsequent task execution through a collaborative propose-and-revise progress. Second, each agent maintains a dynamic internal world model with beliefs. This belief world model is updated via reasoning to infer the intents of collaborators from partial observation and predict the possible states of external world.

To summarize, this work makes the following contributions:

- We propose ***CoBel-World***, a novel framework that integrates a collaborative belief world into LLM agents, enabling efficient communication and consistent planning.
- We design a ***symbolic belief language*** to represent the world knowledge in a structured and explicit form to guide collaboration. We further design a ***Bayesian belief collaboration*** protocol in a Bayesian filter manner, demonstrating how to leverage LLM reasoning capabilities to predict possible beliefs and detect potential miscoordination in a zero-shot manner.
- We evaluate our method on challenging embodied collaboration benchmarks (Zhang et al., 2023) under partial observation. Results show that CoBel-World reduces communication cost by average ***22–60%*** while improving task completion efficiency by average ***4-28%*** on TDW-MAT and C-WAH), outperforming state-of-the-art baseline methods and demonstrating the efficacy of belief-driven collaboration.

## 2 RELATED WORKS

**LLM-Based Multi-Agent Collaboration and Communication.** Recent advances in large language models (LLMs) have enabled their deployment as autonomous agents capable of reasoning, planning, and communication in collaborative settings. Systems such as MetaGPT (Hong et al., 2023) and ChatDev (Qian et al., 2023) demonstrate that LLM agents can follow predefined workflows to solve complex tasks. In embodied intelligence, frameworks like CoELA (Zhang et al., 2023), Capo (Liu et al., 2024b), and RoCo (Mandi et al., 2024) integrate LLMs with perception and action modules to support collaborative embodied tasks. However, these approaches typically rely on fixed communication protocols, such as tep-by-step message generation (Zhang et al., 2023), event-driven multi-round discussion (Liu et al., 2024b), or dense discussion (Guo et al., 2024), leading to excessive communication overhead and poor scalability under partial observability. In contrast, our work introduces a belief-driven communication mechanism that enables LLM agents to dynamically identify and exchange only the most valuable information, significantly reducing communication redundancy while improving collaboration efficiency.

**Belief Modeling in Decentralized Multi-Agent Systems.** In decentralized partially observable Markov decision processes (DEC-POMDP), belief modeling is central to enabling agents to maintain and update probabilistic estimates over hidden states and other agents' intents (Kominis & Geffner, 2015; Moreno et al., 2021). Techniques such as Bayesian reasoning (Foerster et al., 2019) and probabilistic recursive reasoning (Wen et al., 2019) allow agents to infer unobserved variables and align policies through belief estimation. More recent approaches leverage pretrained belief models (Zhai et al., 2023; Pritz & Leung, 2025), achieving improved collaboration in cooperative games such as Hanabi and Overcooked. Wu et al. (2020) leverages inverse planning to infer collaborators' beliefs, allowing agents to dynamically decide between labor division and collaboration. Jha et al. (2024) enables agents to perform higher-order belief modeling with significantly reduced computational cost. Cao et al. (2024) incorporates logical rules to infer human goals and beliefs from demonstrations, thereby guiding hierarchical human–AI collaboration. Despite their success, these methods are largely limited to low-dimensional, discrete-state environments with handcrafted features or require extensive training data. Our work bridges this gap by leveraging the zero-shot

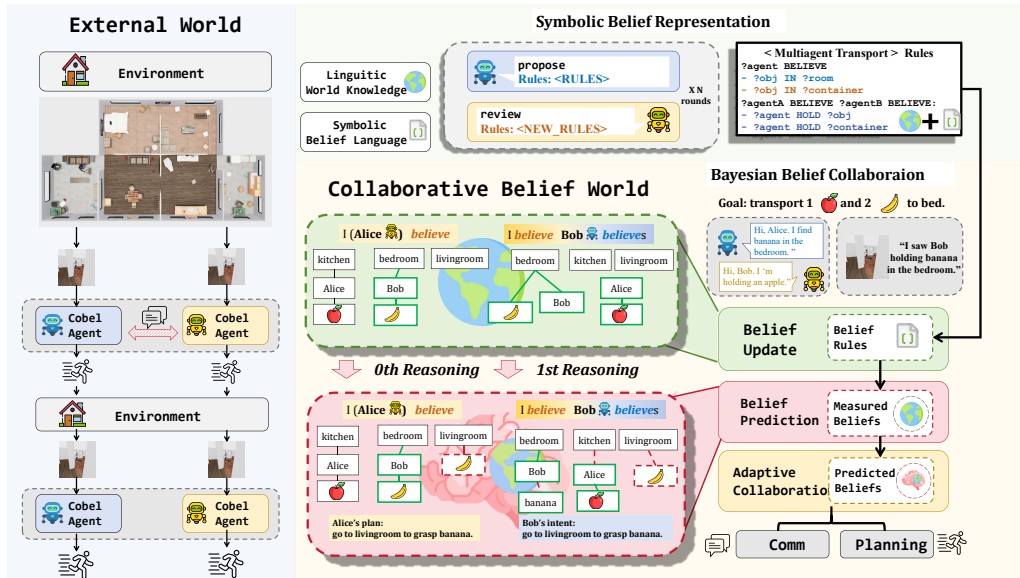

Figure 2: **Overview of Collaborative Belief World framework.** Cobel-World comprises two key components: (1) **Symbolic Belief Representation**: All agents are organized in a collaborative reasoning process to analyze and summarize the rules and requirements of the task in a structured format. With these rules, each agent constructs an initial belief about the world and collaborators; (2) **Bayesian Belief Collaboration**: After the belief world is constructed, each agent updates it via two ways: **belief prediction** (via LLM reasoning) and **belief update** (via observation). Adaptive collaborative decisions will be made based on the beliefs.

reasoning capabilities of LLMs to construct and update structured belief representations in high-dimensional, open-ended physical environments without environment-specific training or explicit state factorization. Recent works (Yi et al., 2025; Zhang et al., 2024) attempt to incorporate belief modeling into LLM-based multi-agent systems to guide decision and strategy selection. However, these works primarily operate under communication-free settings, which limits their scalability in real-world partially observable environments. In contrast, CoBel-World leverages structured belief modeling to guide communication behaviors. Agents with such collaborative belief world can proactively determine when to communicate, whom to communicate with, and how to communicate.

**Reasoning Capabilities in LLM-Based Agents.** The effectiveness of LLMs as autonomous agents hinges on their ability to perform diverse forms of reasoning, from task planning to social inference. Recent work has demonstrated that structured reasoning paradigms significantly enhance agent performance in complex tasks. Notable works include Chain-of-Thought (CoT) (Wei et al., 2022) and Tree of Thoughts (ToT) (Yao et al., 2023), which introduces multi-step reasoning to solve complex problems. More recently, research has advanced social reasoning, particularly theory of mind (ToM), enabling agents to model others' beliefs, intents, and internal states (He et al., 2023; Sclar et al., 2023; Jin et al., 2024; Shi et al., 2024). Several works (Li et al., 2023; Ma et al., 2023; Zhang et al., 2025), have gained benefits in collaborative multi-agent tasks with the introduction of such ability.

## 3 FORMULATION

We model the embodied multi-agent collaboration task as a *decentralized partially observable Markov decision process (DEC-POMDP)* (Oliehoek & Amato, 2016; Bernstein et al., 2000; Spaan et al., 2006a), defined by the tuple:

$$\mathcal{M} = \langle I, \mathcal{S}, \{\mathcal{A}_i\}, \{\mathcal{O}_i\}, T, \{O_i\}, R, h \rangle,$$

where:

- $I = \{1, \ldots, n\}$ is a finite set of $n$ agents;

- $\mathcal{S}$ is a finite state space representing the environment;

- $\mathcal{A}_i$ is the action set of agent $i$, with $\mathcal{A} = \times_{i \in I} \mathcal{A}_i$ the joint action space;

- $\mathcal{O}_i$ is the observation set of agent $i$, encompassing partial egocentric visual inputs and received messages;

- $T(s' \mid s, \mathbf{a}) = p(s' \mid s, \mathbf{a})$ is the transition function, denoting the probability of transitioning to state $s' \in \mathcal{S}$ from $s \in \mathcal{S}$ under joint action $\mathbf{a} \in \mathcal{A}$;

- $O_i(o_i \mid s', \mathbf{a}) = p(o_i \mid s', \mathbf{a})$ is the observation model for agent $i$, giving the probability of observing $o_i \in \mathcal{O}_i$ upon reaching $s'$ after executing $\mathbf{a}$;

- $R(s, \mathbf{a})$ is the global reward function shared by all agents;

- $h$ is the finite planning horizon.

The objective is for the team to maximize the expected cumulative reward $\mathbb{E}\left[\sum_{t=0}^{h-1} R(s^t, \mathbf{a}^t)\right]$ through decentralized execution of a joint policy $\pi = \{\pi_i\}_{i \in I}$, where each agent $i$ selects actions $a_i^t \sim \pi_i(\cdot \mid \tau_i^t)$ based only on its local observation-action history $\tau_i^t = (o_i^0, a_i^0, \ldots, o_i^t)$.

## 4 METHODOLOGY

In this section, we present how CoBel-World leverages belief modeling to address communication redundancy and inconsistent collaboration in embodied multi-agent systems. The theoretical foundation of CoBel-World can be found in Appendix B. Following the paradigm of belief modeling in traditional MARL, we decompose the construction of CoBel-World framework into two components: **Symbolic Belief Representation** for belief representation and **Bayesian Belief Collaboration** for belief update, as depicted in Figure 2.

First, Symbolic Belief Representation (detailed in Section 4.1), centered on a symbolic belief language, enables agents to autonomously interpret task requirements in open-ended environments and encode world knowledge into structured belief rules. It further incorporates a collaborative reasoning process to establish a *collaborative belief world*, ensuring consistent modeling of the environment and collaborators' intents. Second, Bayesian Belief Collaboration (detailed in Section 4.2) maintains and dynamically updates the established belief world during task execution. Agents perform belief updates via a Bayesian filtering scheme powered by LLM reasoning to detect potential miscoordination. When belief misalignment arise, agents proactively communicate to align beliefs and share intents; when beliefs are synchronized, they proceed with action planning and execution. This adaptive mechanism enables context-aware collaboration decisions based on task progress and collaborators' evolving states.

### 4.1 SYMBOLIC BELIEF REPRESENTATION

**Symbolic belief language definition.** LLMs struggle to accurately model diverse and structured beliefs due to the complexity of real-world environments. To address this, we introduce a symbolic belief language inspired by classical planning language (Fox & Long, 2003; Fabiano et al., 2021). We formalize beliefs as tuples consisting of entities, attributes, and predicates. In particular, since beliefs are inherently higher-order (e.g., "Bob believes that Alice believes the apple is in the living room"), we explicitly introduce a recursive belief predicate BELIEVE to capture the collaborators' mental states. The definition of *symbolic belief language* is as follows:

An atomic belief takes one of the following two forms:

```
?belief ::= ?entity PREDICATE ?entity | ?entity ATTRIBUTE ?state
```

where:

- PREDICATE: A relational verb or spatial/state descriptor (e.g., IN, HOLD, AT, INSIDE, NEAR).

- ATTRIBUTE: A unary property of an entity (e.g., EXPLORATION_STATE, CONTENTS).

- `?entity`: A placeholder for any agent, object, or location (e.g., `Alice`, `<apple>`, `<kitchen>`).
- `?state`: A discrete condition or status (e.g., `none`, `part`, `all`, `opened`, `closed`).

The zero-order belief and first-order belief takes the following forms:

- Zero-order belief: `?agent BELIEVE ?belief`.
- First-order belief: `?agentA BELIEVE ?agentB BELIEVE ?belief`.

As illustrated in Figure 2, when Alice observes Bob holding a banana, this visual input is encoded as a zero-order belief: `Alice BELIEVE Bob HOLD <banana>`. When Alice receives the message "I found an apple in the kitchen" from Bob, it is interpreted as a first-order belief: `Alice BELIEVE Bob BELIEVE <apple> IN kitchen`. We provide a detailed description of how unstructured natural language is converted into structured belief representations in Appendix D.1.

**Collaborative representing process.** As shown in Figure 2, we propose a propose-and-revise collaborative representing progress to mitigate hallucination and compositional reasoning failures inherent in LLMs. In this progress, agents iteratively propose and revise the structured belief rules including task constraints, agent capabilities, and logical dependencies. The output of this collaborative progress is a consensus set of belief rules, which constitute a common collaborative belief world and are then used to guide subsequent task execution.

## 4.2 BAYESIAN BELIEF COLLABORATION

In DEC-POMDPs, belief modeling typically follows a Bayesian filtering framework: a **update** that incorporates posterior observation, followed by a **prediction** step based on prior beliefs. We adopt this well-grounded mathematical structure. In the update phase, we generate the agent's beliefs using its partial observation from the environment. In the prediction phase, we leverage the reasoning capabilities of LLMs to predict the potential states of external environment and infer collaborator's intents. The specific design is as follows.

**Belief update.** This step captures the agent's ability to update its beliefs in response to partial observation. We decompose observation into two modalities:

- **Visual observation**: The ego visual perception. (e.g., object positions, agent states);
- **Communication observation**: Messages explicitly transmitted by other agents.

Given the belief rules summarized in the first phase, the agent extracts task-relevant information and updates both **zero-order beliefs** and **first-order beliefs**. Notably, during the update of first-order beliefs, we employ theory-of-mind (ToM) reasoning (Li et al., 2023; Ma et al., 2023) to prompt the agent to interpret messages from the collaborator's perspective. This prevents the agent from conflating personal information with public information, ensuring a more accurate belief estimation. The prompt structure is illustrated below:

---
**Belief Update Prompt**

Prompt: `<Instruct Head>` + `<Partial Observation>` + `<Belief Rules>`
LLM: `<Updated Beliefs>`

---

**Belief prediction.** Building upon the agent's collaborative belief world, we enable proactive coordination by predicting the possible beliefs based on the updated beliefs. Agents perform belief prediction separately based on zero-order and first-order beliefs. For zero-order beliefs, we prompt the LLM to infer possible states of environment. Based on these predicted beliefs, agents then generate plans that maximize task efficiency by prioritizing high-utility, low-uncertainty exploration or manipulation steps. For first-order beliefs, we repeat the reasoning step. However, to ensure comprehensive coverage of potential miscoordination, agent will explicitly reasons over multiple intents for every collaborator—not just the most likely one. This multi-hypothesis modeling allows the agents to fully assess the current collaboration status, guiding their subsequent collaboration behaviors. The prompt structure is illustrated below:

> **Belief Prediction Prompt**
>
> First-order Belief Prediction: `<Instruct Head>` + `<first-order Beliefs>`
> LLM: `<Predicted Beliefs>` + `<Collaborator's Intents>`
> Zero-order Belief Prediction: `<Instruct Head>` + `<zero-order Beliefs>`
> LLM: `<Predicted Beliefs>` + `<My plan>`

**Adaptive collaboration.** After updating and predicting the collaborative belief world, each agent obtains an estimation about collaborators' intents and their mental states, enabling agents to proactively evaluate the current collaboration status. With this capability for dynamic intent inference and state estimation, agents can autonomously and adaptively decide how to collaborate: when potential miscoordination (e.g. conflicting plans) is detected, they send context-aware messages to promote consensus and consistent planning among collaborators; when the current collaboration status is unlikely to cause serious conflicts, agents prefer executing actions directly to improve overall efficiency. To be specific, we first prompt the LLM to explicitly reason over two key aspects: (1) belief misalignment (e.g., Only Bob knows the apple's location.), and (2) potentially conflicting actions (e.g., Alice and Bob plan to explore the same room.). Second, if agents detect the potential miscoordination, they construct a message with the misaligned beliefs and share their intents. Based on this reasoning analysis, agents autonomously adjust their collaboration behaviors, thus promoting efficient, adaptive, and intent-aware collaboration. Details are illustrated in the Figure 2.

## 5 EXPERIMENTS

In this section, we instantiate CoBel-World with diverse LLMs to validate its effectiveness across different benchmarks. First, we compare CoBel-World against several important baselines to demonstrate its superiority in both collaboration efficiency and communication cost. Second, we visualize task trajectories and interaction content to illustrate how CoBel-World leverages belief modeling to facilitate consistent planning and effective communication. Next, we conduct ablation studies to verify the effectiveness of individual modules and extend CoBel-World to scenarios involving more agents to validate its scalability in many-agent environments.

### 5.1 EXPERIMENT SETTINGS

**Benchmarks.** Recently, several benchmarks have been developed to evaluate LLM-based multi-agent systems in embodied environments. PARTNR (Chang et al., 2024) provides a large-scale suite of household tasks to evaluate the reasoning and planning capabilities of LLM-based multi-agent systems. CoELA (Zhang et al., 2023) introduces multiple embodied multi-agent tasks with explicit inter-agent communication channel. To demonstrate CoBel-World's efficiency in communication, we follow CoELA (Zhang et al., 2023) and adopt the two challenging embodied multi-agent benchmarks for our experiments: ThreeDworld Multi-Agent Transport (TDW-MAT) (Zhang et al., 2023), and the Communicative Watch-And-Help (C-WAH) (Zhang et al., 2023). TDW-MAT is built on the general purpose virtual world simulation platform TDW (Gan et al., 2020), and requires agents to move objects by their hands or containers which can contain several objects for efficient moving to the destination. Moreover, agents can receive ego-centric RGB-D images as observation and communicate with others. The test set of TDW-MAT consists of 24 episodes, evenly divided into two task categories: food and stuff. Within each category, episodes are further divided by difficulty into high-capacity (with more containers can be used) and low-capacity settings. In C-WAH, agents are requested to complete five types of household activities, represented as various predicates with specific counts that must be satisfied. The test set contains 10 episodes, including both symbolic and visual observation settings. More details about TDW-MAT and C-WAH environments are provided in Appendix C.1 and C.2, respectively.

**Metrics.** Our evaluation metrics span two dimensions: task completion efficiency and communication cost. For task completion efficiency, we use different metrics for the two benchmarks. On TDW-MAT, we adopt *transport rates* as the primary performance metric, which refers to the fraction of subtasks successfully completed within 3,000 time steps (frames). Note that a single action step may span multiple time steps (e.g., arm resetting). On C-WAH, we report the *average steps* required to complete all tasks, which reflects the efficiency of collaborative coordination. For communication

Table 1: Performance comparison using different LLMs on TDW_MAT benchmark. "↑/↓" means higher/lower is better. Values highlighted in pink denote the best performance, while values underlined indicate the second-best results.

| Task Category | Classic Agents | | Qwen3-32B Agents | | | GPT-4o Agents | | |
|---|---|---|---|---|---|---|---|---|
| | RHP | RHP | CoELA | Capo | CoBel-World | CoELA | Capo | CoBel-World |
| *Transport Rate (↑)* | | | | | | | | |
| Food-low-capacity | 46.67 | 78.33 | 63.33 | 63.33 | 65.00 | 86.67 | 88.33 | **88.33** |
| Stuff-low-capacity | 43.33 | 73.33 | 70.00 | 66.67 | 71.67 | 81.67 | 83.33 | **88.33** |
| **Low-capacity Average** | 45.00 | 75.83 | 66.67 | 65.00 | 68.34 | 84.17 | 85.83 | **88.33** |
| Food-high-capacity | 53.33 | 81.67 | 75.00 | 68.33 | 76.67 | 81.67 | 80.00 | **91.67** |
| Stuff-high-capacity | 50.00 | 65.00 | 58.33 | 71.67 | 58.33 | 71.67 | 78.33 | **78.33** |
| **High-capacity Average** | 51.67 | 73.34 | 66.67 | 70.00 | 67.50 | 76.67 | 79.17 | **85.00** |
| **Total Average** | 48.34 | 74.59 | 66.67 | 67.5 | 67.92 | 80.42 | 82.50 | **86.67** |
| *Communication Cost (↓)* | | | | | | | | |
| Food-low-capacity | — | — | 3549 | 8199 | 2053 | 2117 | 6878 | **1874** |
| Stuff-low-capacity | — | — | 4397 | 7620 | 2092 | 2122 | 7256 | **1506** |
| **Low-capacity-Average** | — | — | 3973 | 7910 | 2073 | 2120 | 7067 | **1690** |
| Food-high-capacity | — | — | 3819 | 7954 | 2103 | 2425 | 5989 | **1786** |
| Stuff-high-capacity | — | — | 3408 | 7395 | 2369 | 2229 | 8178 | **1776** |
| **High-capacity Average** | — | — | 3613 | 7509 | 2236 | 2327 | 6814 | **1781** |
| **Total Average** | — | — | 3793 | 7709 | 2155 | 2224 | 6940 | **1736** |

cost, we compute *the average number of tokens* generated by all agents per episode for communication. Higher transport rates, fewer average steps, and fewer tokens indicate better performance.

**Baselines.** We select two types of baselines for performance comparison: traditional LLM-free agents and LLM-based agents. The traditional agents include: (i) MCTS-based Hierarchical Planner (MHP) (Zhang et al., 2023): A hierarchical planning approach designed for the original Watch-And-Help Challenge. It features a Monte Carlo Tree Search (MCTS)-based high-level planner and a regression-based low-level planner. (ii) Rule-based Hierarchical Planner (RHP) (Zhang et al., 2023): A heuristic-based hierarchical planning approach designed for the original ThreeDWorld Transport Challenge. It uses a rule-based high-level planner combined with an A-start-based low-level planner for navigation. The LLM-based baselines include: (iii) CoELA (Zhang et al., 2023): A collaboration framework based on step-by-step templated message generation and planning. (iv) CaPo (Liu et al., 2024b): A collaboration framework based on event-driven multi-round discussions.

**Implementation details.** To evaluate CoBel-World across different underlying LLMs, we instantiate the LLM-based agents in CoBel-World and other LLM-based baselines using two state-of-the-art models: Qwen3-32B (Yang et al., 2025), an open-source model accessed via the Aliyun API, and ChatGPT-4o (Hurst et al., 2024), a closed-source model accessed via the OpenAI API. We set the parameters with temperature = 0.7, top-p = 1, and a maximum token limit of 512 for both models. Unless otherwise stated, all experiments involve two agents on both benchmarks.

## 5.2 RESULTS

**Performance.** Table 1 and Table 2 compares the performance of different methods on the C-WAH and TDW-MAT benchmarks, respectively. In general, LLM-based agents driven by the small Qwen3-32B perform worse than traditional baselines due to the limited LLM model scale, but agents powered by the more powerful GPT-4o consistently outperform traditional baselines across all test settings. Among them, our CoBel-World framework achieves superior task efficiency over all baseline methods while significantly reducing communication costs. On TDW-MAT, CoBel-World improves average transport rate by **4%** over the best baseline results; on C-WAH, it reduces average steps by **24-28%** compared to the strongest baseline. In terms of communication cost, CoBel-World reduces token usage by **22–60%** across all test settings. These results indicate that belief-driven collaboration not only minimizes redundant communication but also enhances collaboration consistency and planning efficiency. By comparison, baselines such as CoELA and Capo rely on fixed communication protocols to exchange known information and thereby often fail to detect potential miscoordination until conflicting actions occur, leading to the drop of task completion efficiency. Moreover, they initiate communication even when collaboration is unnecessary (e.g., when agents can independently transport all objects in different rooms), causing higher communication costs.

Table 2: Performance comparison using different LLMs on C-WAH benchmark. "↑/↓" means higher/lower is better. Values highlighted in pink denote the best performance, while values underlined indicate the second-best results.

| Task | Obs Type | Classic Agents | | Qwen3-32B Agents | | | GPT-4o Agents | | |
|---|---|---|---|---|---|---|---|---|---|
| | | MHP | MHP | CoELA | Capo | CoBel-World | CoELA | Capo | CoBel-World |
| *Average Step* (↓) | | | | | | | | | |
| Prepare tea | Symbolic Obs | 163 | 87 | 91 | 101 | 106 | 82 | 85 | **53** |
| | Visual Obs | 206 | 102 | 181 | 180 | 105 | 130 | 184 | **91** |
| Wash dishes | Symbolic Obs | 106 | 70 | 48 | 56 | 49 | 46 | 68 | **38** |
| | Visual Obs | 111 | 96 | 95 | 187 | 101 | 76 | 75 | **64** |
| Prepare meal | Symbolic Obs | 105 | 69 | 66 | 87 | 56 | 68 | 66 | **49** |
| | Visual Obs | 181 | 95 | 97 | 151 | 97 | 100 | 83 | **65** |
| Put groceries | Symbolic Obs | 113 | 64 | 82 | 70 | 82 | 64 | 67 | **59** |
| | Visual Obs | 166 | 80 | 108 | 168 | 64 | 82 | 93 | **57** |
| Set up table | Symbolic Obs | 83 | 48 | 69 | 65 | 65 | 56 | 45 | **44** |
| | Visual Obs | 95 | 79 | 115 | 140 | 97 | 102 | 78 | **75** |
| **Symbolic Average** | | 114 | 68 | 71 | 76 | 72 | 63 | 66 | **48** |
| **Visual Average** | | 152 | 90 | 119 | 165 | 93 | 98 | 103 | **71** |
| *Communication Cost* (↓) | | | | | | | | | |
| Prepare tea | Symbolic Obs | — | — | 1114 | 5214 | **386** | 995 | 7027 | 409 |
| | Visual Obs | — | — | 2025 | 5088 | 409 | 964 | 6207 | **399** |
| Wash dishes | Symbolic Obs | — | — | 1095 | 5435 | 332 | 642 | 5587 | **250** |
| | Visual Obs | — | — | 914 | 3708 | 349 | 704 | 4412 | **322** |
| Prepare meal | Symbolic Obs | — | — | 1392 | 9183 | 464 | 1188 | 10244 | **365** |
| | Visual Obs | — | — | 1734 | 5930 | 497 | 1001 | 6028 | **341** |
| Put groceries | Symbolic Obs | — | — | 1124 | 4349 | 453 | 878 | 7163 | **428** |
| | Visual Obs | — | — | 1352 | 4797 | 397 | 862 | 4285 | **395** |
| Set up table | Symbolic Obs | — | — | 1430 | 3671 | **379** | 988 | 6136 | 434 |
| | Visual Obs | — | — | 1242 | 2705 | 430 | 913 | 2547 | **347** |
| **Symbolic Average** | | — | — | 1231 | 5570 | 403 | 938 | 7231 | **377** |
| **Visual Average** | | — | — | 1453 | 4445 | 416 | 889 | 4696 | **360** |

Figure 3: **Illustration of the advantages of CoBel-World in terms of planning consistency and communication efficiency on C-WAH benchmark.** All methods are powered by GPT-4o. The left part illustrates CoBel-World's superior planning consistency over CoELA, while the right panel highlights its reduced communication costs compared to Capo.

**Qualitative analysis.** Figure 3 illustrates the advantages of CoBel-World over baselines in terms of collaboration consistency and communication efficiency. As shown in Figure 3 (left), at the initial stage of the task, agents will first plan their subsequent actions. CoELA follows a fixed pipeline of communication-then-planning, which often fails to reach consensus with collaborators and leads to conflicting plans (e.g., both Alice and Bob intend to explore the same room). In contrast, CoBel-World performs belief prediction before decision-making to reason about the collaborators' intents,

detect potential miscoordination, and proactively initiate communication to reach consensus. For instance, Bob infers that Alice might also explore his current room and thus proactively shares his intent and beliefs with her, enabling more consistent planning. Capo relies on event-triggered multi-round discussions to reach consensus with collaborators. However, when the triggering event provides little or no benefit to collaboration, this mechanism incurs unnecessary communication costs. As illustrated in Figure 3 (right), Capo's discussions fail to yield better plans, resulting in redundant communication. In contrast, CoBel-World leverages belief modeling to autonomously assess the expected utility of communication and dynamically decides whether to communicate to enhance collaboration or directly execute a plan to maximize task efficiency.

## 5.3 ABLATION STUDY

**Effects of each component in CoBel-World.** With C-WAH benchmark as example, we analyze the contributions of two key components in Cobel-World to collaboration: Symbolic Belief Representation (SBP) and Bayesian Belief Collaboration (BBC). As shown in Table 3, after removing the SBR module, Cobel-World exhibits a slight performance drop. This indicates that representing beliefs using unstructured natural language introduces more redundant information, impairing LLMs' planning and decision-making capabilities. In contrast, removing the BBC module leads to a severe performance drop. This phenomenon demonstrates that inferring collaborators' intents significantly enhances agents' ability to perceive the collaborative status and thus enable more context-aware, proactive collaboration.

**Cobel-World with many agents.** Table 4 reports CoBel-World's performance on the C-WAH benchmark as the number of agents scales beyond two. A significant performance gain is observed when scaling from two to three agents. However, increasing the agent number to four yields only marginal improvements in Average Steps. This is because the C-WAH benchmark includes a number of relatively simple tasks composed of only 2–3 subgoals and thus cannot fully leverage the capacity of four agents. As the "wash dishes" task illustrated in the Appendix C.2, only two objects require collection and transport, making collaboration among more than two agents unnecessary and potentially hinder consistent planning.

Table 3: Effects of the components in CoBel-World using GPT-4o on C-WAH benchmark. Average steps required to complete task are reported. "SBP" denotes "Symbolic Belief Representation" and "BBC" denotes "Bayesian Belief Collaboration".

Table 4: Benefits of increasing agent number in our CoBel-World using GPT-4o on C-WAH benchmark. Average steps required for task completion are reported.

| Method | Symbolic Obs (↓) |
|---|---|
| CoBel-World | **51** |
| CoBel-World (No SBR) | 55 |
| CoBel-World (No BBC) | 68 |

| Method | Symbolic Obs (↓) |
|---|---|
| CoBel-World×2 | 51 |
| CoBel-World×3 | 47 |
| CoBel-World×4 | **43** |

## 6 CONCLUDING REMARKS

In this work, we introduced CoBel-World, a framework that equips LLM-based embodied agents with a *collaborative belief world* to enable efficient and consistent multi-agent collaboration under partial observability. CoBel-World formalizes world and mental state knowledge into a structured symbolic belief language and leverages LLMs' zero-shot reasoning for Bayesian-style belief updates. With CoBel-World, LLM agents can proactively infer teammates' intentions and detect potential miscoordination. This intent-aware belief modeling supports adaptive communication, generating messages only when necessary to resolve conflicts or align critical information, thereby reducing redundant dialogue and physical actions. Extensive experiments on challenging benchmarks (TDW-MAT and C-WAH) show that CoBel-World reduces communication costs by 22–60% while consistently improving task completion efficiency over state-of-the-art baselines. These results validate that explicit belief representation is a key enabler of scalable and human-like collaboration in open-ended environments.

## 7 ETHICS STATEMENT

This work involves simulated embodied agents in controlled virtual environments (TDW-MAT and C-WAH) and does not include human subjects, real-world data collection, or deployment in safety-critical settings. All experiments comply with standard research integrity practices. The proposed CoBel-World framework aims to improve communication efficiency and collaboration efficiency among AI agents, with no intent or mechanism to generate harmful, discriminatory, or privacy-invasive behaviors. No external funding sources or conflicts of interest influenced the design or interpretation of this research.

## 8 REPRODUCIBILITY STATEMENT

We have taken multiple steps to ensure the reproducibility of our results. Full experimental details, including environment specifications (TDW-MAT and C-WAH), observation/action spaces, evaluation metrics, and hyperparameters, are provided in Sections 5.1 and Appendix C. The symbolic belief language syntax, belief update/prediction prompts, and Bayesian adaptive collaboration are explicitly defined in Section 4 and Appendix D. Ablation studies and scaling analyses are reported in Section 5.3. While we cannot release code due to double-blind review constraints, all algorithmic components are described with sufficient detail to enable independent reimplementation.

We provide an anonymous github repo with codes for anyone to reproduce CoBel-World. The anonymous github repo url: `https://anonymous.4open.science/r/CoBel_World`

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

# A ADDITIONAL EXPERIMENTS AND ANALYSIS

## A.1 STABILITY EXPERIMENTS

Due to the inherent stochasticity of large language models (LLMs), we conducted multiple evaluation runs to assess the stability of CoBel-World. Specifically, we performed three independent runs on the TDW-MAT benchmark using GPT-4o with a temperature setting of 0.7 and oracle perception. As shown in the Table 5, the results exhibit only minor variance across runs, indicating consistent and reproducible performance.

Table 5: Transport Rate (TR) comparison on the TDW-MAT task using GPT-4o and oracle perception. We perform 3 runs and report mean and variance.

| Runs | Food ($\uparrow$) | Stuff ($\uparrow$) | Avg. ($\uparrow$) |
|---|---|---|---|
| 1 | 0.87 | 0.83 | 0.85 |
| 2 | 0.89 | 0.84 | 0.87 |
| 3 | 0.91 | 0.83 | 0.87 |
| Average | 0.89 (0.016) | 0.83 (0.004) | 0.86 (0.009) |

## A.2 FAILURE ANALYSIS

The failures from CoBel-World primarily stem from inherent hallucinations in LLMs. Although our symbolic belief language substantially reduces such hallucinations, it cannot fully eliminate them. We give detailed failure cases in Figure 4.

## A.3 PERFORMANCE-COST TRADE-OFF REPORT

We report the performance-cost trade-off of CoBel-World and all baselines on the TDW-MAT benchmark in Figure 5. Compared with prior methods, CoBel-World achieves the best trade-off between performance and cost.

# B THEORETICAL ANALYSIS OF COBEL-WORLD

**Belief update with Bayesian filter.** Due to partial observability, each agent $i$ maintains a *belief state* $b_i : \mathcal{S} \to [0, 1]$, which represents its subjective probability distribution over the true state $s \in \mathcal{S}$. The belief $b_i^t$ at time $t$ is conditioned on the agent's local history $\tau_i^t = (o_i^0, a_i^0, \ldots, o_i^t)$. Upon executing action $a^t \in \mathcal{A}$ and receiving observation $o_i^{t+1} \in \mathcal{O}_i$, agent $i$ updates its belief using a Bayesian filter:

$$b_i'(s') \propto O_i(o_i' \mid s', \mathbf{a}) \sum_{s \in \mathcal{S}} T(s' \mid s, \mathbf{a}) \, b_i(s), \tag{1}$$

where $b_i$ is the current belief, $b_i'$ is the updated belief, $\mathbf{a}$ is the joint action, $o_i'$ is the new observation, and $T$ and $O_i$ are the transition and observation models, respectively. This update decomposes into two conceptually distinct stages:

**Prediction:** The agent predict possible beliefs based on its current belief:

$$\bar{b}_i(s') = \sum_{s \in \mathcal{S}} T(s' \mid s, \mathbf{a}) \, b_i(s),$$

resulting in a prior belief $\bar{b}_i$ that captures the expected state distribution after the action. In our framework, this step is enhanced by *theory of mind reasoning* (Li et al., 2023; Ma et al., 2023), enabling agents to anticipate teammates' intentions.

**Measurement update:** The agent conditions the prior on the new observation $o_i'$ (including visual input and messages):

$$b_i'(s') \propto O_i(o_i' \mid s', \mathbf{a}) \cdot \bar{b}_i(s'),$$

**Failure Cases**

***Failure type:*** belief update errors
***Failure log:***
```
Alice: low-level-plan: send a message: Hi Bob, I've
explored part of the <Office>(3000). My subplan is to
continue exploring the <Office>(3000) and transport objects
to the bed.
belief_episode_0 - INFO - first belief update:
first-order beliefs:
Bob BELIEVE Alice BELIEVE <bed> IN <Office> (3000)
```
***Analysis:*** In this case, LLM mistakenly associated `bed` and `office (3000)` in a sentence as belief, and finally got the wrong belief.

***Failure type:*** belief prediction errors
***Failure log:***
```
Bob predict_first: The <Kitchen> (5000) is commonly
associated with food items like oranges, breads, and bananas.
Exploring this room might yield a higher chance of finding
the goal objects.
Bob oppo_subplans:
- go to <Kitchen>(5000)            - explore current room
<Kitchen>(5000)
```
***Analysis:*** LLM mistakenly used common sense as a priori knowledge to aid belief prediction, but there is no setting in the experimental setting that "food is more likely to be in the kitchen", as a result, LLM suggest go to kitchen first.

***Failure type:*** action-selection errors
***Failure log:***
```
Alice intuitive_planning: The last time you saw **Bob was
in the <Office> (1000)**, which hasn't been explored yet. It
may contain target objects or provide useful information for
the goal.
Answer: A. go to <Office> (1000)
```
***Analysis:*** Although LLM believed that Bob is in the `<office>(1000)`, LLM still chose to go to the `<office>(1000)` which is a redundant action.

***Failure type:*** communication content generation error
***Failure log:***
```
Bob: low-level-plan:send a message: Hi Alice, I've explored
all of the <Kitchen> (5000) and the <Livingroom> (8000),
finding <orange> (556740) there. I'm holding <loaf_bread>
(12849971). My subplan is to put <loaf_bread> (12849971)
into <orange> (556740), grasp <orange> (556740), and
transport to the bed.
```
***Analysis:*** put <loaf_bread> (12849971) into <orange> (556740) is inconsistent with common sense and the simulation environment ( Items are randomly initialized in each room) , which is the hallucination of LLM.

Figure 4: Failure cases of CoBel-World.

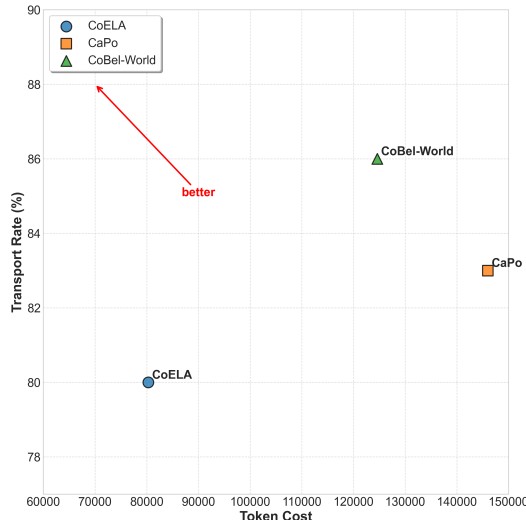

Figure 5: Performance-cost trade-off of CoBel-World and all baselines. The performance metric corresponds to the *Transport Rate* and the cost metric corresponds to the *Token Cost*.

yielding a posterior belief $b_i'$ that incorporates direct evidence. This step enables rapid belief alignment through perception and communication.

This Bayesian-style process—predicting future states and update based on observations—forms the theoretical foundation of our ***CoBel-World*** framework.

## C  ADDITIONAL ENVIRONMENT DETAILS

We evaluate our methods and baseline methods on two challenging embodied multi-agent benchmarks: ThreeDWorld Multi-Agent Transport (TDW-MAT) and Communicative Watch-And-Help (C-WAH). We follow CoELA(Zhang et al., 2023) and the detailed descriptions of these benchmarks are provided below.

### C.1  THREEDWORLD MULTI-AGENT TRANSPORT

**Tasks.**  TDW-MAT comprises two distinct task categories: food-transportation and object-transportation. The food-transportation task involves 6 types of target objects including apple, banana, orange, bread, loaf bread, burger; and three container types: bowl, plate, and tea tray. And the object-transportation task includes another 6 different target objects including calculator, mouse, pen, lighter, purse, iPhone; and three container types: plastic basket, wooden basket, and wicker basket. In each task instance, the environment contains 10 target objects and between 2 to 5 containers. The scenes are structured across four semantically coherent room types: living room, office, kitchen, and bedroom with object placements adhering to real-world contextual plausibility. Agents are required to maximize the number of target objects delivered to a designated goal location within a time budget of 3,000 simulation frames. Containers serve as transport tools, each capable of carrying up to three objects; in their absence, agents can carry at most two objects simultaneously.

**Observation Space.**  The embodied agent receives a variety of observations, with the primary ones being an egocentric RGB image and a depth image. Additionally, there are several auxiliary observations. The observation space includes:

- **RGB image:** An egocentric image captured by a forward-facing camera, with a resolution of $512 \times 512$ and a field of view of 90 degrees.
- **Depth image:** This image shares the same camera intrinsic parameters as the RGB image.
- **Oracle Perception (optional):** An image where each object ID is represented by a distinct color, using the same camera intrinsic parameters as the RGB image.

- **Agent position and rotation:** The position and rotation of the agent within the simulation environment.

- **Messages:** Communications sent by all agents.

- **Held objects:** Information about the objects currently held by the agent.

- **Opponent held objects:** Information about objects held by another agent, if the agent is within view.

**Action Space** In TDW-MAT, agents can perform 7 distinct types of actions to interact with the environment or communicate with each other. Each action spans multiple frames, and the detailed action space is outlined below:

- **Move forward:** The agent advances by 0.5m.

- **Turn left:** The agent rotates left by 15 degrees.

- **Turn right:** The agent rotates right by 15 degrees.

- **Grasp:** The agent grasps an object, successfully performing this action only when in close proximity to the object. The object can be either a target or a container.

- **Put In:** The agent places a target into a container, an action that is possible only when the agent is holding a target in one hand and a container in the other.

- **Drop:** The agent releases the objects held in hand.

- **Send message:** The agent sends a message to others, with a limit of 500 characters per frame.

Table 6: TDW_MAT tasks extended with capacity dimension

| Task Type | Container Num | Container Name |
|---|---|---|
| Food-low-capacity | 2 | tea tray, bowl, plate |
| Food-high-capacity | 5 | tea tray, bowl, plate |
| Stuff-low-capacity | 2 | wood basket, wicker basket, plastic basket |
| Stuff-high-capacity | 5 | wood basket, wicker basket, plastic basket |

**Extended TDW-MAT Tasks.** Building upon the classic TDW-MAT benchmark introduced by CoELA (Zhang et al., 2023), we extend the evaluation along task difficulty dimension to enable a more comprehensive comparison between CoBel-World and various baselines. Specifically, tasks are categorized into low-capacity and high-capacity settings based on the number of containers available to the agent in the environment. Each difficulty level comprises half of both the food-transportation and stuff-transportation tasks. Task details are provided in Table 6.

## C.2 COMMUNICATIVE WATCH-AND-HELP

Communicative Watch-And-Help (C-WAH) builds upon the Watch-And-Help challengeby incorporating the ability for agents to send messages to one another. Sending messages, like other actions, consumes one timestep and is subject to a maximum length constraint.

Table 7: Detailed description of C-WAH tasks

| Task Name | Oject Set |
|---|---|
| Prepare afternoon tea | ON(cupcake,coffeetable), ON(pudding,coffeetable), ON(apple,coffeetable), ON(juice,coffeetable), ON(wine,coffeetable) |
| Wash dishes | IN(plate,dishwasher), IN(fork,dishwasher) |
| Prepare a meal | ON(coffeepot,dinnertable),ON(cupcake,dinnertable), ON(pancake,dinnertable), ON(poundcake,dinnertable), ON(pudding,dinnertable), ON(apple,dinnertable), ON(juice,dinnertable), ON(wine,dinnertable) |
| Put groceries | IN(cupcake,fridge), IN(pancake,fridge), IN(poundcake,fridge), IN(pudding,fridge), IN(apple,fridge), IN(juice,fridge), IN(wine,fridge) |
| Set up a dinner table | ON(plate,dinnertable), ON(fork,dinnertable) |

**Task** The Communicative Watch-And-Help (C-WAH) framework comprises five household-oriented tasks: Prepare afternoon tea, Wash dishes, Prepare a meal, Put groceries, and Set up a dinner table. Each task involves multiple subtasks, expressed through predicates in the form "ON/IN(x, y)", which correspond to actions like "Place x ON/IN y". Some detailed information is provided in Table 7. The primary objective is to complete all given subtasks within 250 timesteps, with each task containing between 3 to 5 subtasks.

**Observation Space** The C-WAH framework provides two observation modalities: Symbolic Observation and Visual Observation. In Symbolic Observation—consistent with the original Watch-And-Help setup—the agent has full access to all object-related information in the same room, including each object's name, location, state, and relational attributes. In Visual Observation, agents receive egocentric RGB and depth images along with auxiliary observations. Detailed observations include:

- **RGB image:** An egocentric image from a forward-facing camera, with a resolution of 256 × 512 and a field of view of 60 degrees.
- **Depth image:** An image with the same camera intrinsic parameters as the RGB image.
- **Oracle Perception:** An image where each object ID is mapped to a color, sharing the same camera intrinsic parameters as the RGB image.
- **Agent position:** The agent's position within the simulation world.
- **Messages:** Communications sent by all agents.
- **Held objects:** Information about the objects currently held by the agent.
- **Opponent held objects:** Information about objects held by another agent, if visible.

**Action Space** The action space in C-WAH closely resembles that of the original Watch-And-Help Challenge, with the addition of the send message action. The detailed action space includes:

- **Walk towards:** Move towards an object in the same room or towards a specific room.
- **Turn left:** Rotate left by 30 degrees.
- **Turn right:** Rotate right by 30 degrees.
- **Grasp:** Grasp an object, which can be successfully performed only when the agent is close to the object.
- **Open:** Open a closed container, performable only when the agent is near the container.
- **Close:** Close an open container, performable only when the agent is near the container.
- **Put:** Place held objects into an open container or onto a surface, performable only when the agent is near the target position.
- **Send message:** Communicate with others, with a limit of 500 characters per message.

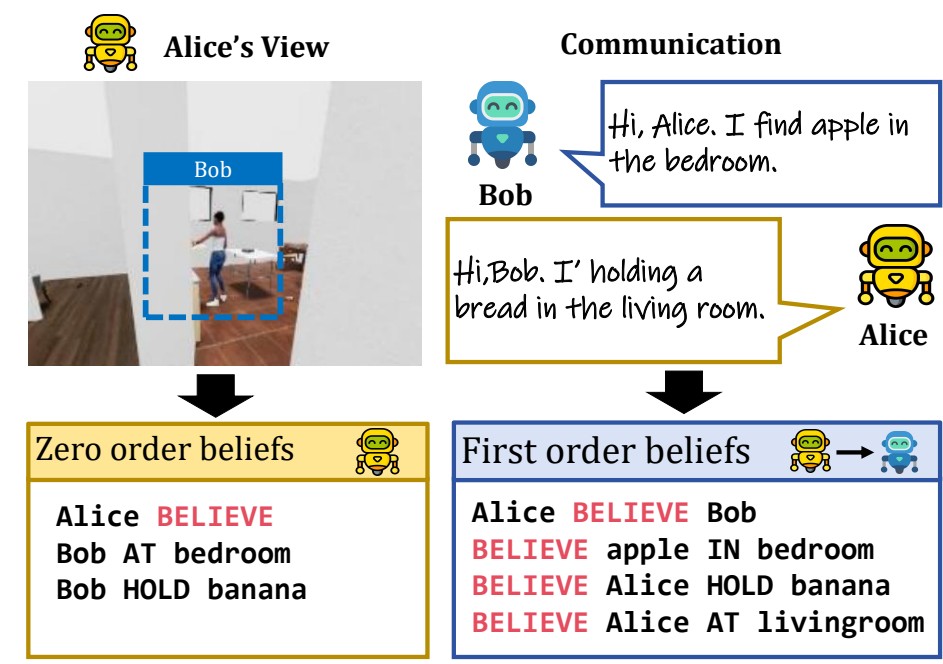

Figure 6: Examples of the transformation from unstructured natural language to structured beliefs.

## D COBEL-WORLD DETAILS

### D.1 BELIEF SYMBOLIC REPRESENTATION

**Examples of Representing Beliefs.** As illustrated in Figure 6, we provide several examples to demonstrate how natural language dialogues and partial visual observations are transformed into structured belief representations.

**Prompt Templates.** We list the belief rules construction prompts for the two agents Alice and Bob in the benchmarks, as shown in Figure 7 and Figure 8, respectively.

**Belief Rules.** Figure 9 illustrates the belief rules of CoBel-World.

### D.2 BAYESIAN BELIEF COLLABORATION

In this part, we list the prompts used in the Bayesian Belief Collaboration module on TDW-MAT benchmark. Figure 11 and Figure 13 illustrate the prompts for zero-order belief update and prediction, respectively. Figure 10 and Figure 12 illustrate the prompts for first-order belief update and prediction, respectively. Figure 14, Figure 15, Figure 16 and Figure 17 depict the prompts for adaptive collaboration, communication, planning and replanning, respectively.

## E LLM USAGE DISCLOSURE

We hereby disclose the use of LLM in the preparation of this manuscript, in compliance with ICLR's submission policies. The LLM was utilized as an assistive tool for language expression refinement during the writing process. Specifically, we leveraged the LLM to optimize the clarity, grammatical accuracy, and writing style of our manuscript. The LLM did not participate in any aspect of research ideation, experimental design or data analysis. All content processed with LLM assistance has undergone thorough review, verification, and manual revision by the authors to ensure scientific accuracy, originality, and consistency with the research findings. We confirm that no content generated by the LLM constitutes plagiarism, fabrication of facts, or other forms of scientific misconduct.

**Belief Rules Construction Prompt of Alice**

**Init Prompt:** You are Alice, you and Bob are constructing beliefs rules to denote the zero and first order belief of the world. You should first extract entity types and predicates in a specific domain given a task description and the belief symbolic language below. After that you should use the belief symbolic language to describe the possible belief types in this task domain and send to bob for discussion.

Belief symbolic language: $BELIEF_LANGUAGE $

Task description: $TASK_DESCRIPTION$
Note that the zeroth-order belief denote my knowledge of the world, first-order belief denote my knowledge of others belief.
DO NOT generate beliefs that go beyond the information specified in the task description.
Consider ONLY zero-order and first-order beliefs.
The belief rules should be in syntax format with entity represented with a "?" prefix, and without any additional comment and analysis and explanation: You should output strictly in the format of the following structure:

Entity and predicate reasoning:
Zero order belief rules:
First order belief rules:

**Refine Prompt:** You are Alice, you and Bob are constructing beliefs rules to denote the zero and first order belief of the world. Given a task description and the belief symbolic language below, you should refine the belief rules according to Bob's suggestions.

Belief symbolic language: $BELIEF_LANGUAGE$
Task description: $TASK_DESCRIPTION$
previous content: $PREVIOUS_CONTENT$
Bob's suggestions: $SUGGESTIONS$

DO NOT generate beliefs that go beyond the information specified in the task description.
Consider ONLY zeroth-order and first-order beliefs.
Note that the zeroth-order belief denote my knowledge of the world, first-order belief denote my knowledge of others belief.
Now try to refine your previous output according to Bob's suggestions. The belief rules should be in syntax formatwith entity represented with a ? prefix, and without any additional comment and analysis and explanation: You should output strictly in the format of the following structure:

Reasoning:
Zero order belief rules:
First order belief rules:

Figure 7: Alice's belief rules construction prompt

**Belief Rules Construction Prompt of Bob**

**Discuss Prompt:** You are bob, you and Alice are constructing belief rules to denote the zero and first order belief of the world. You are required to check the belief rules made by Alice given the challenge description below. Give your reasoning progress in the reasoning:. And then give your comments: Satisfied or Unsatisfied. If Unsatisfied, you should give your suggestions to Alice on how to refine the construction.

These suggestions may include:
Missing logical relationships among key beliefs, such as omitting the agent's belief about its position.
Formatting errors, failing to comply with the prescribed format of the belief language.

Belief symbolic language: $BELIEF_LANGUAGE$ Task description: $TASK_DESCRIP-TION$ Alice content: $ALICE_CONTENT$ Check if Alice's construction satisfy the need. Make deletion advice when occurring repeat syntagma. DO NOT provide suggestions that go beyond the information specified in the task description.
Consider ONLY zeroth-order and first-order beliefs.
Note that the zeroth-order belief denote my knowledge of the world, first-order belief denote my knowledge of others belief.

You should output strictly in the format of the following structure:

Reasoning:
Suggestions:
Satisfied:(yes or no)

Figure 8: Bob's belief rules construction prompt

**Belief Rules**

zero-order belief rules:
```
?agent BELIEVE ?object IN ?room
?agent BELIEVE ?bed IN ?room
?agent BELIEVE ?container IN ?room
?agent BELIEVE ?agent HOLD ?object
?agent BELIEVE ?agent HOLD ?container
?agent BELIEVE ?container CONTAIN ?object
?agent BELIEVE ?room EXPLORED ?exploration_state
?agent BELIEVE ?agent AT ?room
```

first-order belief rules:
```
?agentA BELIEVE ?agentB BELIEVE ?object IN ?room
?agentA BELIEVE ?agentB BELIEVE ?bed IN ?room
?agentA BELIEVE ?agentB BELIEVE ?container IN ?room
?agentA BELIEVE ?agentB BELIEVE ?agent HOLD ?object
?agentA BELIEVE ?agentB BELIEVE ?agent HOLD ?container
?agentA BELIEVE ?agentB BELIEVE ?container CONTAIN ?object
?agentA BELIEVE ?agentB BELIEVE ?room EXPLORED ?exploration_-
state
?agentA BELIEVE ?agentB BELIEVE ?agent AT ?room
```

Figure 9: Illustration of belief rules.

**Prompt for First-order Beliefs Update**

I am $AGENT_NAME$. My teammate $OPPO_NAME$ and I want to transport as many target objects as possible to the bed with the help of containers within 3000 steps.

You are an expert in multi-agent theory-of-mind reasoning. Your task is to analyze, from the perspective of $AGENT_NAME$, what $AGENT_NAME$ believes about what $OPPO_NAME$ knows after the dialogue concludes. This constitutes $AGENT_NAME$'s first-order belief about $OPPO_NAME$'s knowledge.

Given the dialogue history and belief rules below, perform the following steps:
1. Information Extraction Identify all information that $AGENT_NAME$ can infer from the dialogue, including:
Statements made by others to $AGENT_NAME$,
Statements $AGENT_NAME$ themselves made (which reflect their prior knowledge).

2. First-Order Belief Representation:
Based solely on the above information and the provided belief rules, generate $AGENT_-NAME$'s first-order beliefs about $OPPO_NAME$'s knowledge. Notice:
- Adhere strictly to the belief rules; do not introduce external assumptions.
- Replace all placeholders prefixed with "?" with concrete entities mentioned in the dialogue.
- Represent all non-agent entities as <name> (id), e.g., <table> (712).
- Distinguish between private and shared knowledge.
- Beliefs must be expressed in the formal belief rules format—no natural language explanations.

3. Plan Extraction:
Extract $OPPO_NAME$'s explicitly stated or unambiguously expressed plan for their next action, as conveyed in the dialogue. A "plan" refers to an intended future action declared by $OPPO_NAME$. Only include plans that are directly mentioned or clearly articulated; do not infer, complete, or hypothesize based on partial or implicit cues. If no such plan is present, state "None".

Constraints:
Do not generate any information not explicitly present or logically entailed by the dialogue.
All output must conform to the structure and syntax of the belief rules.
Following are provided information for you:

Dialogue History: $MESSAGE$
Belief Rules: $RULE$
Output Format: Extracted Information: [about what $AGENT_NAME$ knows]
First order beliefs: [first-order beliefs in belief rule format]
$OPPO_NAME$'s plan: [concise description of the next plan]

Figure 10: Prompt for the update of first-order beliefs.

**Prompt for Zero-Order Beliefs Update**

I am $AGENT_NAME$. My teammate $OPPO_NAME$ and I want to transport as many target objects as possible to the bed with the help of containers within 3000 steps.

You are an expert in multi-agent theory-of-mind reasoning. Your task is to analyze, from the perspective of $AGENT_NAME$, what $AGENT_NAME$ knows after the dialogue concludes—this includes information conveyed to $AGENT_NAME$ by others, as well as information $AGENT_NAME$ themselves expressed (which reflects their prior knowledge). This constitutes $AGENT_NAME$'s zero-order belief about collaborator's knowledge and task information.

Perform the following steps:

1. Information Extraction:
Extract all information that $AGENT_NAME$ possesses after the dialogue, based solely on the dialogue content.

2. Zero-Order Belief Generation:
Using only the extracted information and the belief rules below, generate $AGENT_-NAME$'s zero-order beliefs.
Notice:
- Adhere strictly to the belief rules; do not introduce external assumptions.
- Replace all placeholders prefixed with "?" with concrete entities mentioned in the dialogue.
- Represent all non-agent entities as <name> (id), such as <table> (712).
- The exploration state of rooms MUST be exactly one of: part, all, or none.
- Beliefs must be expressed exclusively in the formal belief rule format—no natural language explanations.

Constraints:
Do not generate any information not explicitly present or logically entailed by the dialogue.
All output must conform to the structure and syntax of the belief rules.
Following are provided information for you:

Dialogue History: $MESSAGE$

Belief Rules: $RULE$

Answer strictly in this format:

Extracted Information: [about what $AGENT_NAME$ knows]

Zero order beliefs: [zero-order beliefs in belief rules format]

Figure 11: Prompt for the zero-order belief update.

### Prompt for First-Order Beliefs Prediction

I am $AGENT_NAME$. My teammate $OPPO_NAME$ and I want to transport as many target objects as possible to the bed with the help of containers within 3000 steps.

Your task is to simulate $OPPO_NAME$'s decision-making process in a theory-of-mind reasoning style, which is grounded in first-order beliefs about what I $OPPO_NAME$ knows. This first-order beliefs captures $AGENT_NAME$'s understanding of observations, actions, and knowledge of the environment.

First, perform belief-based reasoning: starting from explicit first-order belief, infer the possible beliefs $OPPO_NAME$ may hold about the locations of untransported target objects. e.g., if $AGENT_NAME$ believes a room has been explored "none", then $OPPO_NAME$ may reasonably believe that untransported target objects are likely present in that room. Provide this reasoning process and its conclusion after "reasoning:". You may list at most three concise belief-based justifications.
Second, based on this belief-based reasoning, generate the top three candidate plans that $OPPO_NAME$ is most likely to execute to maximize transport efficiency. Each plan must satisfy the following:
Composed of 1 to 3 atomic actions selected from the allowed set: 1) 'go to': move to a specified room. 2) 'explore current room <room>(id)': explore current room(is not fully explored) for underlying target objects. 3) 'go grasp': go to grasp a specified target object. 4) 'put': Place an object into a specified container. 5) 'transport': Transport holding objects or containers to the bed and drop them on the bed.

Actions take several steps to finish. It may be costly to go to another room or transport to the bed, use these actions sparingly. I can grasp containers and put objects into them to hold more objects at a time. I can hold two items simultaneously (objects or containers). I may grasp only one container at a time. A container can hold up to three objects, enabling transport of up to four items per trip (three inside the container + one in the other hand). Note that containers are discarded upon delivery to the bed. Room exploration states are "none", "part", or "all".

Notice: All entities (rooms, objects, containers) must be strictly represented as <name>(id), e.g., <livingroom>(1000), <wicker_basket>(5388017).

Following are provided information for you:

Goal: $GOAL$

First-order Beliefs: $FIRST_ORDER_BELIEF$

Answer strictly in this format:
reasoning:
plans:
plan1:
plan2:
plan3:

Figure 12: Prompt for first-order beliefs prediction.

## Prompt for Zero-Order Beliefs Prediction

I am $AGENT_NAME$. My teammate $OPPO_NAME$ and I want to transport as many target objects as possible to the bed with the help of containers within 3000 steps.

Your task is to simulate my ($AGENT_NAME$'s) decision-making process, grounded in my zero-order belief—i.e., my direct knowledge of the environment, including observed room exploration states and located objects.

First, perform belief-based reasoning: starting from my explicit zero-order belief, infer the possible beliefs I may hold about the locations of untransported target objects. For example, if I believe a room has been explored "none", I may reasonably believe that untransported target objects are likely present in that room. Provide this reasoning process and its conclusion after "reasoning:". You may list at most three concise belief-based justifications.

Second, based on this belief-based reasoning, generate the best plan I am most likely to execute to maximize transport efficiency. The plan must satisfy the following:

Composed of 1 to 3 atomic actions selected from the allowed set: 1) 'go to': move to a specified room. 2) 'explore current room <room>(id)': explore current room(is not fully explored) for underlying target objects. 3) 'go grasp': go to grasp a specified target object. 4) 'put': Place an object into a specified container. 5) 'transport': Transport holding objects or containers to the bed and drop them on the bed.

Actions take several steps to finish. It may be costly to go to another room or transport to the bed, use these actions sparingly. I can grasp containers and put objects into them to hold more objects at a time. I can hold two items simultaneously (objects or containers). I may grasp only one container at a time. A container can hold up to three objects, enabling transport of up to four items per trip (three inside the container + one in the other hand). Note that containers are discarded upon delivery to the bed. Room exploration states are "none", "part", or "all".

Notice: All entities (rooms, objects, containers) must be strictly represented as <name>(id), e.g., <livingroom>(1000), <wicker_basket>(5388017).

Following are provided information for you:

Goal: $GOAL$

Zero-order Beliefs: $ZERO_ORDER_BELIEF$

Answer strictly in this format:

reasoning:

plan:

Figure 13: Prompt for zero-order beliefs prediction.

**Prompt for Adaptive Collaboration**

I am $AGENT_NAME$. My teammate $OPPO_NAME$ and I want to transport as many target objects as possible to the bed with the help of containers within 3000 steps. I can hold two things at a time, and they can be objects or containers. I can grasp containers and put objects into them to hold more objects at a time. Note that a container can contain three objects, and will be lost once transported to the bed. The room can be explored none/part/all.

Please answer the following questions:

1.Is there any potential miscoordination between my plan and $OPPO_NAME$'s plan, or between my zero-order beliefs and my first-order beliefs about $OPPO_NAME$? Please analyze the miscoordination in two aspects:: (1) Conflicting plans: where my intended actions and $OPPO_NAME$'s intended actions may conflict in space or resource usage. Such as my plan is go to livingroom and explore it, while $OPPO_NAME$'s plans include go to livingroom and grasp the target object there. This is a conflict because we may explore in the same room at the same time which leads to a waste of time. (2) Belief misalignment: where my zero-order belief (what I know) and my first-order belief about $OPPO_NAME$ (what I believe $OPPO_NAME$ knows) are inconsistent regarding critical environmental states, potentially leading to inefficient or contradictory actions. .
For example, I know <kitchen>(2000) is explored "all", but I believe $OPPO_NAME$ thinks <kitchen>(2000) is explored "none". Consequently, $OPPO_NAME$ might waste steps exploring an already fully explored room. Provide your analysis in at most three concise reasons.
2. If the above analysis reveals heavy miscoordination that would significantly impair task efficiency, answer Yes; otherwise, answer No. Minor or non-actionable belief discrepancies that do not lead to conflicting behavior should be tolerated.
3. If your answer is Yes, list the specific pieces of information that are misaligned between my zero-order belief and my first-order belief about $OPPO_NAME$. Itemize only the facts from my zero-order belief that are in conflict with what I believe $OPPO_NAME$ knows. For example: I know <apple>(12123) has been transported. Do not describe $OPPO_-NAME$'s (believed) state.
4. If there is no heavy miscoordination, just answer NO.
Following are provided information for you:

My zero-order belief: $ZERO_ORDER_BELIEF$
My first-order belief about $OPPO_NAME$: $MY_FIRST_ORDER_BELIEF$
My plan: $MY_PLAN$

$OPPO_NAME$'s plans: $OPPO_PLANs$

Answer in this format:
reasons:
answer:
misaligned information:

Figure 14: Prompt for adaptive collaboration.

> **Prompt for Communication Module**
>
> I am $AGENT_NAME$. My teammate $OPPO_NAME$ and I want to transport as many target objects as possible to the bed with the help of containers within 3000 steps. I can hold two things at a time, and they can be objects or containers. I can grasp containers and put objects into them to hold more objects at a time. Note that a container can contain three objects, and will be lost once transported to the bed.
>
> Please help me generate a concise and clear message to inform $OPPO_NAME$ of the misaligned information i know but he don't know and inform $OPPO_NAME$ of my subplan to achieve our shared goal collaboratively. The message should meet following requirements:
>
> 1.The message has to be concise, reliable, and helpful for assisting $OPPO_NAME$ and me to make an efficient and consistent action plan, and transport as many objects to the bed as possible. Don't generate repetitive messages. 2.The message must strictly contain two parts of contents : 1) information only $AGENT_NAME$ know and 2) my plan
>
> Here is an example of generated massage for you:
>
> Example: Message:Hi $OPPO_NAME$, I' ve explored all of the <kitchen>(2000) and found <apple>(12123) there. I'm holding <banana>(12234). My subplan is to grasp <apple>(12123) and transport holding things to the bed.
>
> Just send what $AGENT_NAME$ know, don't need to send what $OPPO_NAME$ knows. Following are provided information for you:
>
> Misaligned information: $MISALIGNED INFORMATION$
> My subplan: $MY_PLAN$

Figure 15: Prompt for communication module.

**Prompt for Planning Module**

I am $AGENT_NAME$. My teammate $OPPO_NAME$ and I want to transport as many target objects as possible to the bed with the help of containers within 3000 steps. I can hold two things at a time, and they can be objects or containers. I can grasp containers and put objects into them to hold more objects at a time. Actions take several steps to finish. It may be costly to go to another room or transport to the bed, use these actions sparingly.

Assume that you are an expert decision maker. Given our shared goal, my current plan, my previous actions, and my zero-order belief (i.e., my direct knowledge of the environment, including observed objects and room exploration states), please: (1) Analyze whether my current plan has been fully executed based on the previous actions and my zero-order belief; (2) If the plan is complete, respond with "SUBPLAN DONE"; (3) If the plan is not yet complete, select the best available next action from the provided action list to achieve the goal as efficiently as possible.
Note: A container can hold up to three objects and is discarded upon transport to the bed. I can only put objects into a container after I have grasped it. All entities must be denoted as <name>(id), e.g., <table>(712).
Please provide up to three concise reasons to support your answer.
Following are provided information for you:

Goal: $GOAL$
My plan: $MY_PLAN$
Previous action: $PREVIOUS_ACTIONS$
My zero-order beliefs: $ZERO_ORDER_BELIEF$
Action list: $ACTION_LIST$

Answer strictly in this format: reasons: answer:

Figure 16: Prompt for planning module.

---

**Prompt for Replanning Module**

I am $AGENT_NAME$. My teammate $OPPO_NAME$ and I want to transport as many target objects as possible to the bed with the help of containers within 3000 steps.

Your task is to simulate my ($AGENT_NAME$'s) decision-making process, grounded in my zero-order belief—i.e., my direct knowledge of the environment, including observed room exploration states and located objects.

First, perform belief-based reasoning: starting from my explicit zero-order belief, infer the possible beliefs I may hold about the locations of untransported target objects. For example, if I believe a room has been explored "none", I may reasonably believe that untransported target objects are likely present in that room. Provide this reasoning process and its conclusion after "reasoning:". You may list at most three concise belief-based justifications.

Second, based on this belief-based reasoning and $OPPO_NAME$'s plan, generate the best plan I should execute to transport target objects as efficiently as possible while actively avoiding conflicts with $OPPO_NAME$'s actions. The plan should complement $OPPO_-NAME$'s activities to maximize overall team efficiency (e.g., by exploring different rooms or handling distinct objects). The generated plan must satisfy the following:
Composed of 1 to 3 atomic actions selected from the allowed set: 1) 'go to': move to a specified room. 2) 'explore current room <room>(id)': explore current room(is not fully explored) for underlying target objects. 3) 'go grasp': go to grasp a specified target object. 4) 'put': Place an object into a specified container. 5) 'transport': Transport holding objects or containers to the bed and drop them on the bed.

Actions take several steps to finish. It may be costly to go to another room or transport to the bed, use these actions sparingly. I can grasp containers and put objects into them to hold more objects at a time. I can hold two items simultaneously (objects or containers). I may grasp only one container at a time. A container can hold up to three objects, enabling transport of up to four items per trip (three inside the container + one in the other hand). Note that containers are discarded upon delivery to the bed. Room exploration states are "none", "part", or "all".
Notice: All entities (rooms, objects, containers) must be strictly represented as <name>(id), e.g., <livingroom>(1000), <wicker_basket>(5388017).

Following are provided information for you:

Goal: $GOAL$

$OPPO_NAME$'s plan: $OPPO_PLAN$
Zero-order Beliefs: $ZERO_ORDER_BELIEF$

Answer strictly in this format:

reasoning:

plan:

Figure 17: Prompt for replanning module.

