# OpenReview forum: "CoBel-World: Harnessing LLM Reasoning to Build a Collaborative Belief World for Optimizing Embodied Multi-Agent Collaboration"
_ICLR.cc/2026/Conference — ICLR 2026 Conference Withdrawn Submission_

### Official Review · Reviewer_YUZ9 · 2025-10-31

**Soundness:** 3
**Presentation:** 2
**Contribution:** 2
**Rating:** 4
**Confidence:** 4

**Summary:**

The paper introduces the Collaborative Belief World framework, equips LLM agents with an internal representation that jointly models the physical environment and collaborators’ mental states. The paper introduces Symbolic Belief Representation for belief representation and Bayesian Belief Collaboration for belief update. Experiments on two embodied benchmarks show the proposed method reduces communication costs and improves task performance.

**Strengths:**

- The introduction of a symbolic belief representation language is novel and well-motivated.

- The experimental results are good and convincing with the newly introduced communication token cost metric.

- Detailed prompt report and code for reproduction.

**Weaknesses:**

- The presentation could use some improvements.
   - Better to put Symbolic Belief Language Definition in the main paper rather than the Appendix, as it's the grounding of the work. Deferring it makes the method hard to follow.
  - Several typos in the manuscript, could use another round of proofreading. E.g., l132-l133 in the contribution
- The method seems to have a strong assumption of homogeneous agents' cooperation, which does not hold for human-agent cooperation.
  - Coming up with agreed Belief Rules (building common ground) itself is a hard problem in decentralized multi-agent cooperation; current implementation seems to be assuming agents of the same architecture (the belief rule construction phase may not be followed perfectly by other agents of other methods, including humans)
  - There's no experiment with heterogeneous agents or human-agent cooperation.

**Questions:**

- How are the ablation experiments of CoBel-World (No SBR) implemented?

- Since the proposed method introduces additional llm call overhead, how's the efficiency tradeoff regarding llm tokens used?

- Missing discussion of recent work about Belief Modeling in multi-agent systems [1][2][3]

[1] COMBO: Compositional World Models for Embodied Multi-Agent Cooperation
[2] Neural Amortized Inference for Nested Multi-agent Reasoning
[3] Too many cooks: Coordinating multi-agent collaboration through inverse planning

---

> ### Author Response · Authors · 2025-12-02
> **Response to Reviewer YUZ9 - Part 1**
>
> We thank you for the comprehensive valuable feedback. We are encouraged as you find *Cobel-World*'s ***belief symbolic language*** design very novel, and consider the experiments and code to be exceptionally solid.
>
> The concerns you raised are addressed as follows:
>
> **Concern#1 Presentation recommendation**
>
> > ***Weakness1.1:*** "Better to put Symbolic Belief Language Definition in the main paper rather than the Appendix, as it's the grounding of the work. Deferring it makes the method hard to follow."
>
> *Our response:*
>
> We are very grateful for this insightful suggestion. We agree that clarifying this key aspect significantly improves the paper's accessibility, and we have accordingly incorporated a detailed definition of the Symbolic Belief Language into the Method section.
>
> > ***Weakness1.2:*** "Several typos in the manuscript, could use another round of proofreading. E.g., l132-l133 in the contribution."
>
> We are grateful for your attention to detail. We agree that these minor issues affect readability, and the typo has been corrected in the latest version of the paper.
>
> **Concern#2 Homogeneous agents' cooperation.**
>
> > ***Weakness1:*** "The method seems to have a strong assumption of homogeneous agents' cooperation, which does not hold for human-agent cooperation."
>
> We acknowledge the concern regarding decentralized and heterogeneous settings. However, CoBel-World does not strictly assume homogeneity in internal belief-based reasoning, for two reasons:
>
> - **Natural Language Interface:** Our framework utilizes natural language as the communication medium. Even if agents possess heterogeneous internal models, they can collaborate effectively as long as they can interpret the standard natural language messages generated by our protocol.
>
> - **Task-Adaptive Belief Rules:** The belief rules are task-adaptive (aiming to maximize environmental rewards like transport efficiency) rather than agent-specific or human-misaligned. This means the "common ground" is derived from the objective physics of the task, not the internal architecture of the agents.
>
> Therefore, even if we test CoBel-World in homogeneous settings, it does not affect its expansion to scenarios of heterogeneous agent collaboration or human-ai collaboration.
>
> **Concern#3 Ablation study implementation.**
>
> > ***Question1:*** "How are the ablation experiments of CoBel-World (No SBR) implemented?"
>
> We thank the reviewer for the question. SBR refers to the *Symbolic Belief Representation*, described in Method §3.1 and Appendix C.1.  In the "*CoBel-World (w/o SBR)*" ablation, we remove the *Belief Update Module*. Instead of converting observations into structured symbolic beliefs, the system directly feeds raw textual environment descriptions (provided by the simulator) into the belief prediction module.
>
> This comparison tests whether structured belief representation offers advantages over unstructured natural language processing. Our results indicate a 7% performance gain with SBR, confirming that symbolic abstraction significantly enhances robustness in open-ended tasks.

---

> ### Author Response · Authors · 2025-12-02
> **Response to Reviewer YUZ9 - Part 2**
>
> **Concern#4 Efficiency tradeoff of CoBel-World.**
>
> > ***Question2:*** "Since the proposed method introduces additional llm call overhead, how's the efficiency tradeoff regarding llm tokens used?"
>
> *Our response:*
>
> We appreciate your concern regarding overall computational cost. Indeed, as shown in Table R.2, CoBel-World incurs a total cost higher than CoELA but lower than CaPo, and Table 2 further reports module-wise cost breakdown. However, these **extra tokens yield a 4%–28% performance improvement**, yielding a substantially better **performance–cost trade-off**.
>
> At the same time, we would like to clarify that this does not contradict the central motivation of CoBel-World—using belief modeling to enable agents to autonomously decide when and what to communicate, thereby reducing *actual communication cost* in collaborative embodied tasks.
>
>  We believe that adaptive, low-overhead communication is essential for real-world multi-agent collaboration:
>
> - **Addressing Bandwidth Limits:** Prior works in MARL have shown that explicitly learning when to communicate (via gating mechanisms [1] or event-triggered strategies [2]) is critical for real-world deployment. In outdoor or multi-device settings with unstable or constrained communication, our belief-driven adaptation mechanism achieves this goal by automatically determining the criticality of communication based on beliefs, effectively performing a message pruning task analogous to these earlier systems.
> - **Mitigating multi-modal, data-intensive collaboration cost:** Previous work has explored efficient communication of multi-sensor multi-modal data[3-4]. Future multi-agent systems will increasingly exchange heavy multi-modal data. Reducing communication frequency becomes crucial to avoid prohibitive bandwidth and latency costs. Our CoBel-World framework addresses this challenge at a higher conceptual level: our **Adaptive Belief Collaboration** mechanism fundamentally decides whether and how to compress information based on the belief regarding the collaboration utility (i.e., the criticality of the information).
>
> Finally, we emphasize that CoBel-World achieves the **best performance–cost trade-off**(*We provide visualizations in our latest submission Appendix A.3* ) : by leveraging structured belief modeling, it both (i) improves task performance and (ii) allows agents to *minimize communication by actively deciding when communication is necessary*. This aligns directly with the practical requirements of real-world embodied multi-agent cooperation.
>
> **Table R.2. Total token cost (prompt + completion) comparison across methods.**
>
> | Avg token   | CoELA       | CaPo     | CoBel      |
> | ----------- | ----------- | -------- | ---------- |
> | Plan        | **44237.3** | 68904.4  | 118057.5   |
> | Comm        | 36053.1     | 77079.9  | **6523.9** |
> | Total       | **80290.4** | 145984.3 | 124581.4   |
> | Performance | 80          | 83       | **86**     |
>
> [1] Mao H, Zhang Z, Xiao Z, et al. Learning agent communication under limited bandwidth by message pruning[C]//Proceedings of the AAAI Conference on Artificial Intelligence. 2020, 34(04): 5142-5149.
>
> [2] Hu G, Zhu Y, Zhao D, et al. Event-triggered multi-agent reinforcement learning with communication under limited-bandwidth constraint[J]. arXiv preprint arXiv:2010.04978, 2020.
>
> [3] Yang D, Yang K, Wang Y, et al. How2comm: Communication-efficient and collaboration-pragmatic multi-agent perception[J]. Advances in Neural Information Processing Systems, 2023, 36: 25151-25164.
>
> [4] Hu Y, Fang S, Lei Z, et al. Where2comm: Communication-efficient collaborative perception via spatial confidence maps[J]. Advances in neural information processing systems, 2022, 35: 4874-4886.

---

> ### Author Response · Authors · 2025-12-02
> **Response to Reviewer YUZ9 - Part 3**
>
> **Concern#5 The lack of discussion and comparison with relevant prior work**
>
> > ***Question3：*** "Missing discussion of recent work about Belief Modeling in multi-agent systems [1][2][3]"
>
> *Our repsponse:*
>
> We are very grateful for pointing out these relevant references. They were not included in the original submission because **CoBel-World specifically targets proactive belief-driven communication decision**, which differs from the goals of **[1-3]**.
>
> - COMBO [1] learns compositional world models for communication-free coordination using predefined state structures.
>
> - Neural Amortized Inference [2] and Too Many Cooks [3] operate within traditional RL frameworks using low-dimensional, handcrafted belief spaces without explicit communication channels.
>
> In contrast, CoBel-World operates in open-ended, high-dimensional environments and uniquely leverages belief modeling to drive communication decisions, a capability not addressed in these works.
>
> To ensure complete clarity regarding the distinctions between these important contributions and our approach, **we have incorporated this detailed comparison into the Related Work section of the revised manuscript.**
>
> [1] Zhang H, Wang Z, Lyu Q, et al. COMBO: compositional world models for embodied multi-agent cooperation[J]. arXiv preprint arXiv:2404.10775, 2024.
>
> [2] Jha K, Le T A, Jin C, et al. Neural amortized inference for nested multi-agent reasoning[C]//Proceedings of the AAAI Conference on Artificial Intelligence. 2024, 38(1): 530-537.
>
> [3] Wu S A, Wang R E, Evans J A, et al. Too many cooks: Coordinating multi-agent collaboration through inverse planning[C]//Proceedings of the annual meeting of the cognitive science society. 2020, 42.

---

### Official Review · Reviewer_xCs1 · 2025-10-31

**Soundness:** 3
**Presentation:** 2
**Contribution:** 2
**Rating:** 4
**Confidence:** 4

**Summary:**

The authors propose a novel framework, CoBel-World (Collaborative Belief World), to improve decentralized LLM agents collaboration using belief modeling - specifically to produce consistent plans and reduce communication costs.

They create a symbolic belief language to represent mental state, current plan, and collaborator intents for each agent. They also describe first-order belief (i.e. I believe agent B believes..). They perform zero-shot Bayesian belief updates with theory-of-mind reasoning based on observations and communication. Then, the belief prediction for future intents and states is done based on the prior beliefs. The communication only happens when there is a misalignment after the belief update.

The authors use transport rates for TDW-MAT, and average steps to complete all tasks for C-WAH for efficiency calculation. For communication, they use the average number of takes generated by all agents for communication on average.

They deploy Qwen3-32B and ChatGPT-4o models as agents. They compare against traditional (MHP, RHP) as well as other LLM-based frameworks (CoELA, CaPo) as baselines on the TDW-MAT and C-WAH benchmarks. The CoBel-world framework shows significantly reduced communication costs and improved completion efficiency, especially with ChatGPT-4o. They also perform ablations by removing each component - symbolic belief representation, and the bayesian belief collaboration, and show that while both are important, bayesian belief collaboration is essential for high performance for CoBel-world. They also experiment with more agents to show that framework can scale.

**Strengths:**

1. They apply a useful idea from traditional multi-agent RL (belief networks/modeling) to zero-shot MLLM agents and show improved performance and costs on benchmarks.
2. They use actual embodied benchmarks instead of grid-like or text-only worlds, which is closer to the real-world.
3. Lack of task-specific fine-tuning allows the framework to potentially generalize and scale to any environment.

**Weaknesses:**

1. The novelty of the approach is very limited. While the approach works better than baselines, the ideas of belief modeling are not uncommon in multi-agent RL. This is just application of the idea in the LLM-agent setting.
2. The authors do not cite or compare against very easy-to-find, relevant, and perhaps similar works. A simple Google Search leads to many Belief Modeling + Belief Language + LLM ideas: https://openreview.net/pdf?id=TWC4gLoAxY, https://arxiv.org/pdf/2506.08292
3. The benchmarks are very small (24 episodes and 10 episodes) and have limited tasks, which are perhaps not enough to signal a high-performance in the real-world. The benchmarks they use are typically only good to test 2 agents in collaboration, 3 at best.
4. There are many typos in the paper, and missing letters.
	1. Line 132 - averafe -> average
	2. Line 146 - tep-by-step -> step-by-step
	3. Line 224 -> defination -> definition
	4. "Collaborative representing progress" -> do the authors mean process?
	5. Line 293 -> pontential -> potential
	6. Line 501 - Baesian -> Bayesian
	7. Line 505 - reproduct -> reproduce

**Questions:**

1. Can the authors clarify the differences against the recent works, and any other that incorporate belief modeling with LLMs?
2. How can we get a better signal as to whether this approach applies generally to the real-world? Have the authors considered PARTNR: https://ai.meta.com/research/publications/partnr-a-benchmark-for-planning-and-reasoning-in-embodied-multi-agent-tasks/?

---

> ### Author Response · Authors · 2025-12-02
> **Response to Reviewer xCs1 - Part 1**
>
> We thank you for the constructive feedback. We are encouraged that you recognize our *CoBel-World* as effective, training-free and closer to the real-world settings. The concerns you raised are addressed as follows:
>
> **Concern#1 The lack of Novelty**
>
> > ***Weakness1:*** "The novelty of the approach is very limited. While the approach works better than baselines, the ideas of belief modeling are not uncommon in multi-agent RL. This is just application of the idea in the LLM-agent setting."
>
> *Our response:*
>
> Thanks for your comment. Although belief modeling has a long history in multi-agent RL, *CoBel-World is the first to  incorporate belief modeling into LLM-based systems to explicitly optimize inter-agent communication.* *To tackle* *the challenges of this problem, CoBel-World introduces many* *unique design components*.
>
> For instance, it introduces ***Symbolic Belief Language*** that represent unstructured information as structured beliefs to enhance robust reasoning and collaboration . Additionally, *CoBel-World* incorporates ***Adaptive Belief Collaboration*** to enable agents to adaptively select communication or execution.
>
> **Concern#2 Limited experiment settings**
>
> > ***Weakness2.1:*** "The benchmarks they use are typically only good to test 2 agents in collaboration, 3 at best."
> >
> > ***Question2:*** "How can we get a better signal as to whether this approach applies generally to the real-world? Have the authors considered PARTNR?"
>
> *Our response:*
>
> Thank you for raising this valuable concern. Although the number of episodes is limited, each task is long-horizon and partially observable requiring multi-step belief modeling and coordination. Moreover, **both TDW-MAT and CWAH include clear difficulty gradations** (i.e. TDW-MAT varies the number of containers, while CWAH tasks differ in target objects and interaction logic) which provide diverse tasks for evaluating the generality of *CoBel-World*. These tasks are **substantially more demanding than discrete benchmarks**. Our objective is to evaluate the mechanism of LLM-based belief-driven collaboration and communication, and environments such as TDW-MAT and CWAH allow us to isolate and analyze this behavior more precisely.
>
> We also note that most large-scale embodied benchmarks, including PARTNR[1], **do not support explicit inter-agent communication**, which is essential for evaluating our belief-driven communication policy. Therefore, such benchmarks cannot directly assess the core capability introduced by CoBel-World.
>
> We appreciate that PARTNR is a high-quality benchmark with realistic household tasks and a robust infrastructure. We have added a discussion on PARTNR in the *Experiement Settings* section, acknowledging its significance while clarifying why our specific communication-centric focus necessitated the use of TDW-MAT and CWAH.
>
> > ***Weakness2.2:*** "The benchmarks they use are typically only good to test 2 agents in collaboration, 3 at best."
>
> *Our response:*
>
> Regarding the number of agents, CoBel-World already includes experiments with up to four agents (*Table 4 in our submission*), our framework can naturally scale to more agents. We did not include larger-team evaluations mainly because **current embodied benchmarks are not designed for reliable testing beyond four agents**, as noted in *lines 510–514*. Therefore, we adopt the commonly used agent scale for fair and meaningful comparison.
>
> Additionally, the settings with 2–4 agents are the **standard configuration in the embodied multi-agent community**: the works cited in our submission (e.g., CoELA[2], Capo[3], Roco[4] ) and the references[5-7] suggested by Reviewer YUZ9 and MLjZ also use no more than four agents. Thus, our evaluation follows widely adopted community practice.
>
> [1] Chang M, Chhablani G, Clegg A, et al. Partnr: A benchmark for planning and reasoning in embodied multi-agent tasks[J]. arXiv preprint arXiv:2411.00081, 2024.
>
> [2] Zhang H, Du W, Shan J, et al. Building cooperative embodied agents modularly with large language models[J]. arXiv preprint arXiv:2307.02485, 2023.
>
> [3] Liu J, Zhou P, Du Y, et al. CaPo: Cooperative Plan Optimization for Efficient Embodied Multi-Agent Cooperation[J]. arXiv preprint arXiv:2411.04679, 2024.
>
> [4] Mandi Z, Jain S, Song S. Roco: Dialectic multi-robot collaboration with large language models[C]//2024 IEEE International Conference on Robotics and Automation (ICRA). IEEE, 2024: 286-299.
>
> [5] Zhang H, Wang Z, Lyu Q, et al. COMBO: compositional world models for embodied multi-agent cooperation[J]. arXiv preprint arXiv:2404.10775, 2024.
>
> [6] Jha K, Le T A, Jin C, et al. Neural amortized inference for nested multi-agent reasoning[C]//Proceedings of the AAAI Conference on Artificial Intelligence. 2024, 38(1): 530-537.
>
> [7] Wu S A, Wang R E, Evans J A, et al. Too many cooks: Coordinating multi-agent collaboration through inverse planning[C]//Proceedings of the annual meeting of the cognitive science society. 2020, 42.

---

> ### Author Response · Authors · 2025-12-02
> **Response to Reviewer xCs1 - Part 2**
>
> **Concern#3 Typo errors**
>
> > ***Weakness3:*** "There are many typos in the paper, and missing letters."
>
> We are grateful for your attention to detail. We agree that these minor issues affect readability, and the typo has been corrected in the latest version of the paper.
>
> **Concern#4 The lack of discussion and comparison with relevant prior work**
>
> > ***Weakness2:*** "The authors do not cite or compare against very easy-to-find, relevant, and perhaps similar works. A simple Google Search leads to many Belief Modeling + Belief Language + LLM ideas"
> >
> > ***Question1:*** "Can the authors clarify the differences against the recent works, and any other that incorporate belief modeling with LLMs?"
>
> We thank you very much for highlighting these relevant references. They were omitted in the original submission because CoBel-World specifically targets proactive belief-driven communication within embodied closed-loop collaboration, which differs fundamentally from the reasoning-centric tasks in [1] and [2].
>
> *ECON* [1] focus on leveraging beliefs to enhance LLM debate via Bayesian Nash equilibrium, operating in a communication-free setting. In contrast, *CoBel-World* leverages belief modeling not only for collaborative decision-making but also to enable agents to proactively optimize communication decisions (when and whom to message).
>
> [2] utilizes logic-guided Theory of Mind (ToM) to improve intent inference for Human-AI collaboration within an RL framework. Unlike their reliance on predefined logical structures, *CoBel-World* introduces a unified symbolic belief language capable of zero-shot, high-dimensional belief representation, addressing the challenges of open-ended embodied environments.
>
> We have incorporated these works into the ***Related Work (Part 2)*** section of the revised manuscript with an explicit comparison.
>
> [1] Yi X, Zhou Z, Cao C, et al. From Debate to Equilibrium: Belief-Driven Multi-Agent LLM Reasoning via Bayesian Nash Equilibrium[J]. arXiv preprint arXiv:2506.08292, 2025.
>
> [2] Cao C, Fu Y, Xu S, et al. Enhancing human-AI collaboration through logic-guided reasoning[C]//The Twelfth International Conference on Learning Representations. 2024.

---

### Official Review · Reviewer_MLjZ · 2025-11-04

**Soundness:** 3
**Presentation:** 2
**Contribution:** 2
**Rating:** 4
**Confidence:** 4

**Summary:**

The paper introduces CoBel-World, a framework that augments LLM agents with an explicit "collaborative belief world." A symbolic belief language encodes task knowledge and agent mental states. At the same time, a Bayesian-style update loop, with prediction and measurement steps executed via LLM prompting, maintains and refines those beliefs during execution. On two embodied multi-agent benchmarks, compared with rule-based planners and two recent LLM collaboration baselines, CoBel-World reduces communication tokens by 22–60% and yields 4–28% better task-completion efficiency.

**Strengths:**

1. By explicitly distinguishing zero- and first-order beliefs, the authors provide a concrete representation that is easier to inspect than pure free text.

2. The Bayesian filter implemented through LLM prompting is a creative use of reasoning capability that decides whether to speak or act.

3. On both TDW-MAT and C-WAH, CoBel-World achieves the best transport rate / step count while cutting token usage substantially. In Figure 3 the qualitative trajectory shows how belief prediction avoids room-duplication and unnecessary chat.

**Weaknesses:**

1. The “Bayesian” update is performed by textual prompting rather than probabilistic computation, and the normalization and sampling policies are not well-defined.

2. Recent work on ToM reasoning and belief-driven LLM collaboration/debate is not cited or contrasted, such as [1][2]. Differences in representation and update procedure should be spelled out.

3. All results appear to be single-run numbers. No variance or statistical tests are provided. Given the inherent stochasticity of LLM outputs (temperature 0.7), this casts doubt on the reported average gains of 4%.

4. The method may trade fewer tokens between agents for more tokens issued to the LLM internally (belief prediction, misalignment detection). There is no report of total LLM calls or wall-clock time, so efficiency is not well-supported.

5. There are too many grammatical mistakes and typos that affect readability. For example, in the caption for Figure 2: "All agent" should be "All agents", "collaborative reasoning progress" should be "collaborative reasoning process", "to analysis" should be "to analyze", "structured format" should be "a structured format", "each agent construct a initial belief" should be "each agent constructs an initial belief", "each agent update" should be "each agent updates", and "adaptive collaborative decision" should be "an adaptive collaborative decision".

[1] AutoToM: Scaling Model-based Mental Inference via Automated Agent Modeling. Zhang et al, 2025.

[2] From Debate to Equilibrium: Belief-Driven Multi-Agent LLM Reasoning via Bayesian Nash Equilibrium. Yi et al, 2025.

**Questions:**

1. How many internal LLM calls (update + prediction + alignment checks) are issued per environment step compared to CoELA/CaPo? Please report average total tokens (internal + external) and time.

2. Can you conduct some failure analysis to understand when the framework will fail?

---

> ### Author Response · Authors · 2025-12-02
> **Response to Reviewer MLjZ - Part 1**
>
> Thanks for the thoughtful feedback. We are encouraged that you find our paper creative, particularly in its explicit formulation of beliefs and its strong empirical results. Some specific concerns are addressed as follows.
>
> **Concern#1** **Textual-prompting Bayesian update.**
>
> >  ***Weakness1:*** "The "Bayesian" update is performed by textual prompting rather than probabilistic computation, and the normalization and sampling policies are not well-defined.''
>
> *Our response:*
>
> We thank you very much for the insightful comment. *CoBel-World* does not claim to perform exact Bayesian inference. Instead,  we design our method by following the belief-update formulation in the *POMDP* framework, specifically the equation of Bayesian filter:
>
> - Our *belief prediction* step corresponds to the prior *prediction* step of Bayesian filter, and the *belief update* step corresponds to the posterior *measurement update* of Bayesian filter. We give a detailed theoretical analysis in *Appendix B*.
> - Our work shares a underlying design principle with recent works[1-2] in LLM-based Bayesian planning, using mathematical formulations to guide system-level prompting design.
>
> [1] Jin, Chuanyang et al. “MMToM-QA: Multimodal Theory of Mind Question Answering.” *ArXiv* abs/2401.08743 (2024): n. pag.
>
> [2] Zhang Z, Jin C, Jia M Y, et al. Autotom: Scaling model-based mental inference via automated agent modeling[C]//The Thirty-ninth Annual Conference on Neural Information Processing Systems. 2025.
>
> **Concern#2** **The lack of discussion and comparison with relevant prior work**
>
> > ***Weakness2:*** "Recent work on ToM reasoning and belief-driven LLM collaboration/debate is not cited or contrasted, such as [1] [2]. Differences in representation and update procedure should be spelled out."
>
> *Our response:*
>
> We thank you for pointing out these references. These works were not emphasized in our original submission because **CoBel-World focuses on proactive, belief-driven communication for embodied agents in open environments**, which differs from the objectives of the cited papers.
>
> AutoToM [1] focuses on latent mental state inference using Bayesian networks and inverse planning to model the beliefs. However, it does not utilize these beliefs to actively control inter-agent communication protocols.
> ECON [2] frames multi-LLM interactions as a Bayesian game to reach Nash equilibria, aiming to enhance collaborative reasoning in debate-style tasks. It focuses on belief-guided reasoning without direct communication between agents rather than dynamic, zero-shot belief construction and adaptive communication in open-world embodied environments.
>
> We have included explicit citations and detailed comparisons to Autotom (*Related Work part3*) and ECON (*Related Work part2*) in the revised submission.
>
> [1] Zhang Z, Jin C, Jia M Y, et al. Autotom: Scaling model-based mental inference via automated agent modeling[C]//The Thirty-ninth Annual Conference on Neural Information Processing Systems. 2025.
>
> [2] Yi X, Zhou Z, Cao C, et al. From Debate to Equilibrium: Belief-Driven Multi-Agent LLM Reasoning via Bayesian Nash Equilibrium[J]. arXiv preprint arXiv:2506.08292, 2025.
>
>
>
> **Concern#3 Potential stochasticity induced by LLM under temperature 0.7**
>
> > ***Weakness3:*** "All results appear to be single-run numbers. No variance or statistical tests are provided. Given the inherent stochasticity of LLM outputs (temperature 0.7), this casts doubt on the reported average gains of 4%."
>
> We appreciate reviewer’s concern regarding the stochasticity of LLMs. To address this concern, we conducted additional three-run experiments using the same temperature setting (T=0.7)**.  **As shown in Table R.1, CoBel-World exhibits high stability, with performance varying within a narrow margin of **±2%**. The average success rate across runs is **86.33%**, which aligns closely with the **86.67%** reported in our initial submission. This demonstrates that our framework is robust against temperature-induced stochasticity.
>
> We attribute this stability to our ***Symbolic Belief Representation***. By constraining reasoning to a structured symbolic belief space (e.g., object locations, exploration states) rather than free-form text, we significantly reduce the solution space, making the LLM inference more reliable and less sensitive to stochastic variation.
>
> We also added this content as *stability experiments* part in out latest submission Appendix A.1.
>
> **Table R.1. Multi-run performance of CoBel-World on TDW-MAT.**
>
> | Runs    | Food (↑)     | Stuff (↑)    | Avg. (↑)     |
> | :------ | :----------- | :----------- | :----------- |
> | 1       | 0.87         | 0.83         | 0.85         |
> | 2       | 0.89         | 0.84         | 0.87         |
> | 3       | 0.91         | 0.83         | 0.87         |
> | Average | 0.89 (0.016) | 0.83 (0.004) | 0.86 (0.009) |

---

> ### Author Response · Authors · 2025-12-02
> **Response to Reviewer MLjZ - Part 2**
>
> **Concern#4 Total token cost of *CoBel-World***
>
> > ***Weakness4:*** "The method may trade fewer tokens between agents for more tokens issued to the LLM internally (belief prediction, misalignment detection). There is no report of total LLM calls or wall-clock time, so efficiency is not well-supported."
> >
> > ***Question1:*** "How many internal LLM calls (update + prediction + alignment checks) are issued per environment step compared to CoELA/CaPo? Please report average total tokens (internal + external) and time."
>
> *Our response regarding computational cost:*
>
> We appreciate your concern regarding overall computational cost. Indeed, as shown in Table R.2, CoBel-World incurs a total cost higher than CoELA but lower than CaPo, and Table R.2 further reports module-wise cost breakdown. However, these **extra tokens yield a 4%–28% performance improvement**, yielding a substantially better **performance–cost trade-off**.
>
> At the same time, we would like to clarify that this does not contradict the central motivation of CoBel-World—using belief modeling to enable agents to autonomously decide when and what to communicate, thereby reducing *actual communication cost* in collaborative embodied tasks.
>
>  We believe that adaptive, low-overhead communication is essential for real-world multi-agent collaboration:
>
> - **Addressing Bandwidth Limits:** Prior works in MARL have shown that explicitly learning when to communicate (via gating mechanisms [1] or event-triggered strategies [2]) is critical for real-world deployment. In outdoor or multi-device settings with unstable or constrained communication, our belief-driven adaptation mechanism achieves this goal by automatically determining the criticality of communication based on beliefs, effectively performing a message pruning task analogous to these earlier systems.
> - **Mitigating multi-modal, data-intensive collaboration cost:** Previous work has explored efficient communication of multi-sensor multi-modal data[3-4]. Future multi-agent systems will increasingly exchange heavy multi-modal data. Reducing communication frequency becomes crucial to avoid prohibitive bandwidth and latency costs. Our CoBel-World framework addresses this challenge at a higher conceptual level: our **Adaptive Belief Collaboration** mechanism fundamentally decides whether and how to compress information based on the belief regarding the collaboration utility (i.e., the criticality of the information).
>
> Finally, we emphasize that CoBel-World achieves the **best performance–cost trade-off** (*We provide visualizations in our latest submission Appendix A.3* ) : by leveraging structured belief modeling, it both (i) improves task performance and (ii) allows agents to *minimize communication by actively deciding when communication is necessary*. This aligns directly with the practical requirements of real-world embodied multi-agent cooperation.
>
> **Table R.2. Total token cost (prompt + completion) comparison across methods.**
>
> | Avg token   | CoELA       | CaPo     | CoBel    |
> | ----------- | ----------- | -------- | -------- |
> | Plan        | 44237.3     | 68904.4  | 118057.5 |
> | Comm        | 36053.1     | 77079.9  | 6523.9   |
> | Total       | **80290.4** | 145984.3 | 124581.4 |
> | Performance | 80          | 83       | **86**   |
>
> *Our response regarding time:*
>
> In TDW-MAT tasks, episodes have a fixed upper limit of 3000 frames. Agents attempt to complete as many goals as possible within this limit. Simply comparing "Episode Time" is misleading because CoBel-World often completes **more goals** within the same frames.
>
> Instead, we report the **Average Frames Per Goal** in **Table R.3**. CoBel-World demonstrates superior efficiency, requiring significantly fewer frames to achieve each objective.
>
> **Table R.3. Efficiency comparison across different methods.**
>
> | Metric (less is better) | CoELA | CaPo  | CoBel     |
> | ----------------------- | ----- | ----- | --------- |
> | Avg frame per goal      | 366.7 | 358.6 | **334.5** |
>
> [1] Mao H, Zhang Z, Xiao Z, et al. Learning agent communication under limited bandwidth by message pruning[C]//Proceedings of the AAAI Conference on Artificial Intelligence. 2020, 34(04): 5142-5149.
>
> [2] Hu G, Zhu Y, Zhao D, et al. Event-triggered multi-agent reinforcement learning with communication under limited-bandwidth constraint[J]. arXiv preprint arXiv:2010.04978, 2020.
>
> [3] Yang D, Yang K, Wang Y, et al. How2comm: Communication-efficient and collaboration-pragmatic multi-agent perception[J]. Advances in Neural Information Processing Systems, 2023, 36: 25151-25164.
>
> [4] Hu Y, Fang S, Lei Z, et al. Where2comm: Communication-efficient collaborative perception via spatial confidence maps[J]. Advances in neural information processing systems, 2022, 35: 4874-4886.

---

> ### Author Response · Authors · 2025-12-02
> **Response to Reviewer MLjZ - Part 3**
>
> **Concern#5 Typo errors**
>
> > ***Weakness5:*** "There are too many grammatical mistakes and typos that affect readability."
>
> *Our response:*
>
> We are grateful for your attention to detail. We agree that these minor issues affect readability, and the typo has been corrected in the latest version of the paper.
>
>
>
> **Concern#6 Failure cases**
>
> > ***Question2:***  "Can you conduct some failure analysis to understand when the framework will fail?"
>
> *Our response:*
>
> The remaining errors in CoBel-World primarily stem from inherent hallucinations in LLMs. Although our symbolic belief language substantially reduces such hallucinations, it cannot fully eliminate them. These errors mainly manifest in the following forms:
>
> - **Belief update errors**, where the model incorrectly extracts information or produces belief entries that do not conform to the defined symbolic format.
>
>   > Alice: low-level-plan: send a message: Message: Hi Bob, I've explored part of the <Office>(3000). My subplan is to continue exploring the **<Office>(3000)** and transport objects to the **bed**.
>   > 2025-09-15 12:44:18,266 - belief_episode_0 - INFO - first belief update:
>   > first-order beliefs:
>   > **(ERROR BELIEF)** Bob BELIEVE Alice BELIEVE <bed> IN <Office> (3000)
>
>   *Analysis:* In this case, LLM mistakenly associated `bed` and `office (3000)` in a sentence as belief, and finally got the wrong belief.
>
> - **Belief prediction errors**, in which the model generates logically inconsistent or incorrect reasoning contents.
>
>   > Bob predict_first:
>   >
>   > 2. **The <Kitchen> (5000) is commonly associated with food items like oranges, breads, and bananas.** Exploring this room might yield a higher chance of finding the goal objects.
>
>   > Bob oppo_subplans:
>   >
>   > \- go to <Kitchen>(5000)
>   >
>   > \- explore current room <Kitchen>(5000)
>
>    *Analysis:* LLM mistakenly used common sense as a priori knowledge to aid belief prediction, but there is no setting in the experimental setting that "food is more likely to be in the kitchen", as a result, LLM suggest go to kitchen first.
>
> - **Action-selection errors**, where the agent fails to follow the intended sub-plan and instead chooses irrelevant actions.
>
>   > Alice intuitive_planning:
>   >
>   > reasons:
>   >
>   > 1. The subplan is to explore the current room <Livingroom> (7000), and you have already completed exploring it and found nothing relevant to the goal.
>   > 2. Since you haven't found the bed or any target objects in the current room, continuing to explore other rooms is necessary to locate target objects.
>   > 3. The last time you saw **Bob was in the <Office> (1000)**, which hasn't been explored yet. It may contain target objects or provide useful information for the goal.
>   >
>   >  **answer: A. go to <Office> (1000)**
>
>   *Analysis:* Although LLM believed that Bob is in the `<office>(1000)` ,  LLM still choosed to go to the `<office>(1000)` which is a redundant action.
>
> - **Communication content generation error**
>
>   > Bob: low-level-plan:send a message: Hi Alice, I've explored all of the <Kitchen> (5000) and the <Livingroom> (8000), finding <orange> (556740) there. I'm holding <loaf_bread> (12849971). My subplan is to put <loaf_bread> (12849971) into <orange> (556740), grasp <orange> (556740), and transport to the bed.
>
>   *Analysis:* `put <loaf_bread> (12849971) into <orange> (556740)` is inconsistent with common sense and the simulation environment ( Items are randomly initialized in each room) , which is the hallucination of LLM.
>
> We also added this content as *failure analysis* part in out latest submission Appendix A.2.

---

### Note · Authors · 2026-01-06

I have read and agree with the venue's withdrawal policy on behalf of myself and my co-authors.